# Neural Tangent Kernel Empowered Federated Learning

## Abstract

Federated learning (FL) is a privacy-preserving paradigm where multiple participants jointly solve a machine learning problem without sharing raw data. Unlike traditional distributed learning, a unique characteristic of FL is statistical heterogeneity, namely, data distributions across participants are different from each other. Meanwhile, recent advances in the interpretation of neural networks have seen a wide use of neural tangent kernel (NTK) for convergence and generalization analyses. In this paper, we propose a novel FL paradigm empowered by the NTK framework. The proposed paradigm addresses the challenge of statistical heterogeneity by transmitting update data that are more expressive than those of the traditional FL paradigms. Specifically, sample-wise Jacobian matrices, rather than model weights/gradients, are uploaded by participants. The server then constructs an empirical kernel matrix to update a global model without explicitly performing gradient descent. We further develop a variant with improved communication efficiency and enhanced privacy. Numerical results show that the proposed paradigm can achieve the same accuracy while reducing the number of communication rounds by an order of magnitude compared to federated averaging.

## 1 Introduction

Federated learning (FL) has emerged as a popular paradigm involving a large number of workers collaboratively solving a machine learning problem (Kairouz et al., 2021). In a typical FL framework, a server broadcasts a global model to selected workers and collects model updates without needing to access the raw data. One popular algorithm is known as *federated averaging* (FedAvg) (McMahan et al., 2017), in which workers perform stochastic gradient descent (SGD) to update the local models and upload the weight vectors to the server. A new global model is constructed on the server by averaging the received weight vectors.

Li et al. (2020) characterized some unique challenges of FL. First, client data are generated locally and remain decentralized, which implies that they may not be independent and identically distributed (IID). Prior works have shown that statistical heterogeneity can negatively influence the convergence of FedAvg (Zhao et al., 2018). This phenomenon may be explained that local updating under data heterogeneity will cause cost-function inconsistency (Wang et al., 2020). More challengingly, the learning procedure is susceptible to system heterogeneity, including the diversity of hardware, battery power, and network connectivity. Local updating schemes often exacerbate the straggler issue caused by heterogeneous system characteristics.

Recent studies have proposed various strategies to alleviate the statistical heterogeneity. One possible solution is to share a globally available dataset with participants to reduce the distance between client-data distributions and the population distribution (Zhao et al., 2018). In practice, though, such a dataset may be unavailable or too small to meaningfully compensate for the heterogeneity. Some researchers replaced the coordinate-wise weight averaging strategy in FedAvg with nonlinear aggregation schemes (Wang et al., 2020; Chen & Chao, 2021). The nonlinear aggregation relies on a separate optimization routine, which can be elusive, especially when the algorithm does not converge well. Another direction is to modify the local objectives or local update schemes to cancel the effects of client drift (Li et al., 2020; Karimireddy et al., 2020). However, some studies reported that these methods are not consistently effective, and may perform worse than FedAvg when evaluated in various settings (Reddi et al., 2021; Haddadpour et al., 2021; Chen & Chao, 2021).

In this work, we present a neural tangent kernel empowered federated learning (NTK-FL) paradigm. Given a fixed number of communication rounds, NTK-FL outperforms state-of-the-art methods in terms of test accuracy. We summarize our contributions as follows.

- We propose a novel FL paradigm without requiring workers to perform gradient descent. To the best of our knowledge, this is the first work using the NTK method to replace gradient descent in FL algorithms.

- Our scheme inherently solves the non-IID data problem of FL. Compared with FedAvg, it is robust to different degrees of data heterogeneity and has a consistently fast convergence speed. We verify the effectiveness of the paradigm theoretically and experimentally.

- We add **c**ommunication-efficient and **p**rivacy-preserving features to the paradigm and develop CP-NTK-FL by combining strategies such as random projection and data subsampling. We show that some strategies can also be applied to traditional FL methods. Although such methods cause performance degradation when applied to FedAvg, they only slightly worsen the model accuracy when applied to the proposed CP-NTK-FL.

## 2 RELATED WORK

**Neural Tangent Kernel.** Jacot et al. (2018) showed that training an infinitely wide neural network with gradient descent in the parameter space is equivalent to kernel regression in the function space. Lee et al. (2019) used a first-order Taylor expansion to approximate the neural network output and derived the training dynamics in a closed form. Chen et al. (2020) established the generalization bounds for a two-layer over-parameterized neural network with the NTK framework. The NTK computation has been extended to convolutional neural networks (CNNs) (Arora et al., 2019), recurrent neural networks (RNNs) (Alemohammad et al., 2021), and even to neural networks with arbitrary architectures (Yang & Littwin, 2021). Empirical studies have also provided a good understanding of the wide neural networks training (Lee et al., 2020).

**Federated Learning.** FL aims to train a model with distributed workers without transmitting local data (McMahan et al., 2017; Kairouz et al., 2021). FedAvg has been proposed as a generic solution with many theoretical analyses and implementation variants. Recent studies have shown a growing interest in improving its communication efficiency, privacy guarantees, and robustness to heterogeneity. To reduce communication cost, gradient quantization and sparsification were incorporated into FedAvg (Reisizadeh et al., 2020; Sattler et al., 2019). From the security perspective, Zhu et al. (2019) showed that sharing gradients may cause privacy leakage. To address this challenge, differentially private federated optimization and decentralized aggregation methods were developed (Girgis et al., 2021; Cheng et al., 2021). Other works put the focus on the statistical heterogeneity issue and designed various methods such as adding regularization terms to the objective function (Li et al., 2020; Smith et al., 2017). In this work, we focus on a novel FL paradigm where the global model is derived based on the NTK evolution. We show that the proposed NTK-FL is robust to statistical heterogeneity by design, and extend it to a variant with improved communication efficiency and enhanced privacy.

**Kernel Methods in Federated Learning.** The NTK framework has been mostly used for convergence analyses in FL. Seo et al. (2020) studied two knowledge distillation methods in FL and compared their convergence properties based on the neural network function evolution in the NTK regime. Li et al. (2021) incorporated batch normalization layers to local models, and provided theoretical justification for its faster convergence by studying the minimum nonnegative eigenvalue of the tangent kernel matrix. Huang et al. (2021) directly used the NTK framework to analyze the convergence rate and generalization bound of two-layer ReLU neural networks trained with FedAvg. Su et al. (2021) studied the convergence behavior of a set of FL algorithms in the kernel regression setting. In comparison, our work does not focus on pure convergence analyses of existing algorithms. We propose a novel FL framework by replacing the gradient descent with the NTK evolution.

## 3 Background and Preliminaries

We use lowercase nonitalic boldface, nonitalic boldface capital, and italic boldface capital letters to denote vectors, matrices, and tensors, respectively. For example, for column vectors $\mathbf{a}_j \in \mathbb{R}^M$, $j \in \{1, \ldots, N\}$, $\mathbf{A} = [\mathbf{a}_1, \ldots, \mathbf{a}_N]$ is an $M \times N$ matrix. A third-order tensor $\boldsymbol{A} \in \mathbb{R}^{K \times M \times N}$ can be viewed as a concatenation of such matrices. We use a *slice* to denote a matrix in a third-order tensor by varying two indices (Kolda & Bader, 2009). Take tensor $\boldsymbol{A}$, for instance: $\boldsymbol{A}_{i::}$ is a matrix of the $i$th horizontal slice, and $\boldsymbol{A}_{:j:}$ is its $j$th lateral slice (Kolda & Bader, 2009). Finally, the indicator function of an event is denoted by $\mathbb{1}(\cdot)$.

### 3.1 Federated Learning Model

Consider an FL architecture where a server trains a global model by indirectly using datasets distributed among $M$ workers. The local dataset of the $m$th worker is denoted by $\mathcal{D}_m = \{(\mathbf{x}_{m,i}, \mathbf{y}_{m,i})\}_{i=1}^{N_m}$, where $(\mathbf{x}_{m,i}, \mathbf{y}_{m,i})$ is an input-output pair, drawn from a distribution $\mathcal{P}_m$. The local objective can be formulated as an empirical risk minimization over $N_m$ training examples: $F_m(\mathbf{w}) = \frac{1}{N_m} \sum_{i=1}^{N_m} R(\mathbf{w}; \mathbf{x}_{m,j}, \mathbf{y}_{m,i})$, where $R$ is a sample-wise risk function quantifying the error of model with a weight vector $\mathbf{w} \in \mathbb{R}^d$ estimating the label $\mathbf{y}_{m,i}$ for an input $\mathbf{x}_{m,i}$. The global objective function is denoted by $F(\mathbf{w})$, and the optimization problem may be formulated as:

$$\min_{\mathbf{w} \in \mathbb{R}^d} F(\mathbf{w}) = \frac{1}{M} \sum_{m=1}^{M} F_m(\mathbf{w}). \tag{1}$$

### 3.2 Linearized Neural Network Model

Let $(\mathbf{x}_i, \mathbf{y}_i)$ denote a training pair, with $\mathbf{x}_i \in \mathbb{R}^{d_1}$ and $\mathbf{y}_i \in \mathbb{R}^{d_2}$, where $d_1$ is the input dimension and $d_2$ is the output dimension. $\mathbf{X} \triangleq [\mathbf{x}_1, \ldots, \mathbf{x}_N]^\top$ represents the input matrix and $\mathbf{Y} \triangleq [\mathbf{y}_1, \ldots, \mathbf{y}_N]^\top$ represents the label matrix. Consider a neural network function $\mathbf{f} : \mathbb{R}^{d_1} \to \mathbb{R}^{d_2}$ parameterized by a vector $\mathbf{w} \in \mathbb{R}^d$, which is the vectorization of all weights for the multilayer network. Given an input $\mathbf{x}_i$, the network outputs a prediction $\hat{\mathbf{y}}_i = \mathbf{f}(\mathbf{w}; \mathbf{x}_i)$. Let $\ell(\hat{\mathbf{y}}_i, \mathbf{y}_i)$ be the loss function measuring the dissimilarity between the predicted result $\hat{\mathbf{y}}_i$ and the true label $\mathbf{y}_i$. We are interested in finding an optimal weight vector $\mathbf{w}^\star$ that minimizes the empirical loss over $N$ training examples:

$$\mathbf{w}^\star = \operatorname*{argmin}_{\mathbf{w}} L(\mathbf{w}; \mathbf{X}, \mathbf{Y}) \triangleq \frac{1}{N} \sum_{i=1}^{N} \ell(\hat{\mathbf{y}}_i, \mathbf{y}_i). \tag{2}$$

One common optimization method is the gradient descent training. Given the learning rate $\eta$, gradient descent updates the weights at each time step as: $\mathbf{w}^{(t+1)} = \mathbf{w}^{(t)} - \eta \nabla_{\mathbf{w}} L$. To simplify the notation, let $\mathbf{f}^{(t)}(\mathbf{x})$ be the output at time step $t$ with an input $\mathbf{x}$, i.e., $\mathbf{f}^{(t)}(\mathbf{x}) \triangleq \mathbf{f}(\mathbf{w}^{(t)}; \mathbf{x})$. Following Lee et al. (2019), we use the first-order Taylor expansion around the initial weight vector $\mathbf{w}^{(0)}$ to approximate the neural network output given an input $\mathbf{x}$, i.e.,

$$\mathbf{f}^{(t)}(\mathbf{x}) \approx \mathbf{f}^{(0)}(\mathbf{x}) + \mathbf{J}^{(0)}(\mathbf{x})(\mathbf{w}^{(t)} - \mathbf{w}^{(0)}), \tag{3}$$

where $\mathbf{J}^{(0)}(\mathbf{x}) = [\nabla \mathbf{f}_1^{(0)}(\mathbf{x}), \ldots, \nabla \mathbf{f}_{d_2}^{(0)}(\mathbf{x})]^\top$, with $\nabla \mathbf{f}_j^{(t)}(\mathbf{x}) \triangleq [\partial \hat{y}_j^{(t)} / \partial w_1^{(t)}, \ldots, \partial \hat{y}_j^{(t)} / \partial w_d^{(t)}]^\top$ being the gradient of the $j$th component of the neural network output with respect to $\mathbf{w}^{(t)}$. Consider the halved mean-squared error (MSE) loss $\ell$, namely, $\ell = \frac{1}{d_2} \sum_{j=1}^{d_2} \frac{1}{2} (\hat{y}_j - y_j)^2$. Based on the continuous-time limit, one can show that the dynamics of the gradient flow are governed by the following differential equation:

$$\frac{d\mathbf{f}}{dt} = -\eta \mathbf{H}^{(0)} (\mathbf{f}^{(t)}(\mathbf{X}) - \mathbf{Y}), \tag{4}$$

where $\mathbf{f}^{(t)}(\mathbf{X}) \in \mathbb{R}^{N \times d_2}$ is a matrix of concatenated output for all training examples, and $\mathbf{H}^{(0)}$ is the neural tangent kernel at time step 0, with each entry $(\mathbf{H}^{(0)})_{ij}$ equal to the scaled Frobenius inner product of the Jacobian matrices: $(\mathbf{H}^{(0)})_{ij} = \frac{1}{d_2} \langle \mathbf{J}^{(0)}(\mathbf{x}_i), \mathbf{J}^{(0)}(\mathbf{x}_j) \rangle_{\mathrm{F}}$. The differential equation (4) has the closed-form solution:

$$\mathbf{f}^{(t)}(\mathbf{X}) = \left(\mathbf{I} - e^{-\frac{\eta t}{N} \mathbf{H}^{(0)}}\right) \mathbf{Y} + e^{-\frac{\eta t}{N} \mathbf{H}^{(0)}} \mathbf{f}^{(0)}(\mathbf{X}). \tag{5}$$

The neural network state $\mathbf{f}^{(t)}(\mathbf{X})$ can thus be directly obtained from (5) without running the gradient descent algorithm.

## 4    PROPOSED FL PARADIGM VIA THE NTK FRAMEWORK

In this section, we present the NTK-FL paradigm (Figure 1) and then extend it to the variant CP-NTK-FL (Figure 2) with improved communication efficiency and enhanced privacy. The detailed algorithm descriptions are presented as follows.

### 4.1    NTK-FL PARADIGM

NTK-FL follows four steps to update the global model in each round. **First**, the server will select a subset $\mathcal{C}_k$ of workers and broadcast to them a model weight vector $\mathbf{w}^{(k)}$ from the $k$th round. Here, the superscript $k$ is the communication round index, and it should be distinguished from the gradient descent time step $t$ in Section 3.2. **Second**, each worker will use its local training data $\mathcal{D}_m$ to compute a Jacobian tensor $\boldsymbol{J}_m^{(k)} \in \mathbb{R}^{N_m \times d_2 \times d}$, $\forall\ m \in \mathcal{C}_k$, which is a concatenation of $N_m$ sample-wise Jacobian matrices $(\boldsymbol{J}_m^{(k)})_{i::} = [\nabla \mathbf{f}_1^{(k)}(\mathbf{x}_{m,i}), \ldots, \nabla \mathbf{f}_{d_2}^{(k)}(\mathbf{x}_{m,i})]^\top$, $i \in \{1, \ldots, N_m\}$. The worker will then upload the Jacobian tensor $\boldsymbol{J}_m^{(k)}$, labels $\mathbf{Y}_m$, and initial condition $\mathbf{f}^{(k)}(\mathbf{X}_m)$ to the server. The transmitted information corresponds to the variables in the state evolution of $\mathbf{f}^{(t)}$ in (5). **Third**, the server will construct a global Jacobian tensor $\boldsymbol{J}^{(k)} \in \mathbb{R}^{N \times d_2 \times d}$ based on received $\boldsymbol{J}_m^{(k)}$'s, with each worker contributing $N_m$ horizontal slices to $\boldsymbol{J}^{(k)}$.

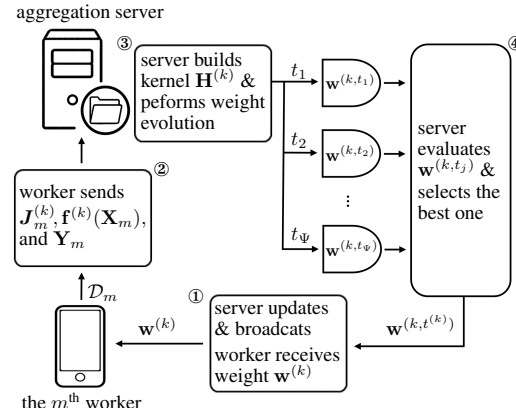

Figure 1: Schematic of NTK-FL. Each worker first receives the weight $\mathbf{w}^{(k)}$, and then uploads the Jacobian tensor $\boldsymbol{J}_m^{(k)}$, label $\mathbf{Y}_m$, and initial condition $\mathbf{f}^{(k)}(\mathbf{X}_m)$. The server builds a global kernel $\mathbf{H}^{(k)}$ and performs the weight evolution with $\{t_1, \ldots, t_\Psi\}$. We use (9) to find the best $t_j$ and update the weight accordingly.

We use a toy example to explain the process as follows. Suppose the server selects worker 1 and worker 2 in a certain round. Workers 1 and 2 will compute the Jacobian tensors $\boldsymbol{J}_1^{(k)}$ and $\boldsymbol{J}_2^{(k)}$, respectively. The global Jacobian tensor is constructed as:

$$\boldsymbol{J}_{i::}^{(k)} = \begin{cases} \boldsymbol{J}_{1,i::}^{(k)} & , \text{ if } i \in \{1, \ldots, N_1\}, \\ \boldsymbol{J}_{2,j::}^{(k)} & , j = i - N_1, \text{ if } i \in \{N_1 + 1, \ldots, N_1 + N_2\}. \end{cases} \tag{6}$$

After obtaining the global Jacobian tensor $\boldsymbol{J}^{(k)}$, the $(i,j)$th entry of the global kernel $\mathbf{H}^{(k)}$ is calculated as the scaled Frobenius inner product of two horizontal slices of $\boldsymbol{J}^{(k)}$, i.e., $(\mathbf{H}^{(k)})_{ij} = \frac{1}{d_2}\langle \boldsymbol{J}_{i::}^{(k)}, \boldsymbol{J}_{j::}^{(k)} \rangle_{\mathrm{F}}$. For simplicity, **Fourth**, the server will perform the NTK evolution to obtain the updated neural network function $\mathbf{f}^{(k+1)}$ and weight vector $\mathbf{w}^{(k+1)}$. With a slight abuse of notation, let $\mathbf{f}^{(k,t)}$ denote the neural network output at gradient descent step $t$ in communication round $k$. The neural network function evolution dynamics and weight evolution dynamics are given by:

$$\mathbf{f}^{(k,t)} = \left( \mathbf{I} - e^{-\frac{\eta t}{N} \mathbf{H}^{(k)}} \right) \mathbf{Y}^{(k)} + e^{-\frac{\eta t}{N} \mathbf{H}^{(k)}} \mathbf{f}^{(k)}, \tag{7a}$$

$$\mathbf{w}^{(k,t)} = \sum_{j=1}^{d_2} (\boldsymbol{J}_{:j:}^{(k)})^\top \mathbf{R}_{:j}^{(k,t)} + \mathbf{w}^{(k)}, \tag{7b}$$

where $\boldsymbol{J}_{:j:}^{(k)}$ is the $j$th lateral slice of $\boldsymbol{J}^{(k)}$, and $\mathbf{R}_{:j}^{(k,t)}$ is the $j$th column of the residual matrix $\mathbf{R}^{(k,t)}$ defined as follows:

$$\mathbf{R}^{(k,t)} \triangleq \frac{\eta}{Nd_2} \sum_{u=0}^{t-1} \left[ \mathbf{Y}^{(k)} - \mathbf{f}^{(k,u)}(\mathbf{X}^{(k)}) \right]. \tag{8}$$

Note that $\mathbf{X}^{(k)} = [\mathbf{X}_1^\top, \ldots, \mathbf{X}_{\mathcal{C}_k}^\top]^\top$ denotes a concatenation of worker training examples, and $\mathbf{Y}^{(k)} = [\mathbf{Y}_1^\top, \ldots, \mathbf{Y}_{\mathcal{C}_k}^\top]^\top$ denotes a concatenation of worker labels. The weight evolution in (7b) is derived by unfolding the gradient descent steps. To update the global weight, the server performs the evolution with various integer steps $\{t_1, \ldots, t_\Psi\}$ and selects the best one with the smallest loss value. Our goal is to minimize the training loss with a small generalization gap (Nakkiran et al., 2020). The updated weight is decided by the following procedure:

$$\mathbf{w}^{(k+1)} \triangleq \mathbf{w}^{(k)}, \quad t^{(k)} = \underset{t_j}{\operatorname{argmin}}\, L(\mathbf{f}(\mathbf{w}^{(k,t_j)}; \mathbf{X}^{(k)}, \mathbf{Y}^{(k)})). \tag{9}$$

Alternatively, if the server has an available validation dataset, the optimal number of update steps can be selected based on the model validation performance. In practice, such a validation dataset can be obtained from held-out workers (Wang et al., 2021). Based on the closed-form solution in (7b), the search of $t^{(k)}$ over the grid $\{t_1, \ldots, t_\Psi\}$ can be completed in $O(\Psi)$ time.

**Comparison of NTK-FL and Huang et al. (2021).** Huang et al. (2021) presented the details of FedAvg by letting clients use local updates and upload gradients to train a two-layer neural network. In contrast, NTK-FL let each client transmit Jacobian matrices without performing local SGD steps. The model weight is updated via NTK evolution in (7b). The main differences include: (1) clients transmit more expressive Jacobian matrices to improve model performance in the non-IID FL setting; (2) the computation is shifted to the server.

**Robustness Against Statistical Heterogeneity.** In essence, statistical heterogeneity comes from the decentralized data of heterogeneous distributions owned by individual workers. If privacy is not an issue, the non-IID challenge can be readily resolved by mixing all workers' datasets and training a centralized model. In NTK-FL, instead of building a centralized dataset, we use Jacobian matrices to construct a global kernel $\mathbf{H}^{(k)}$, which is a concise representation of gathered data points from all selected workers. This representation is yet more expressive/less compact than that of a traditional FL algorithm. More precisely, the update being sent for NTK-FL regarding the $i$th training example of the $m$th worker for NTK-FL is $\mathbf{J}_m = [\nabla \mathbf{f}_1(\mathbf{x}_{m,i}), \ldots, \nabla \mathbf{f}_{d_2}(\mathbf{x}_{m,i})]^\top$, whereas the gradient update being sent for FedAvg is $\nabla L(\mathbf{w}; \mathbf{x}_{m,i}, \mathbf{y}_{m,i}) = \frac{1}{d_2} \sum_{j=1}^{d_2} (\hat{y}_{m,i,j} - y_{m,i,j}) \nabla \mathbf{f}_j(\mathbf{x}_{m,i})$, a weighted sum of coordinates of $\mathbf{J}_m$. By sending Jacobian matrices $\mathbf{J}_m$ and jointly processing them on the server, NTK-FL delays the more aggressive data aggregation step after the communication stage and therefore better approximates the centralized learning setting than FedAvg does.

## 4.2 CP-NTK-FL VARIANT

Compared to FedAvg, NTK-FL does not incur additional client computational overhead since calculating the Jacobian tensor enjoys the same communication efficiency with computing aggregated gradients. Without locally updating weight vectors, NTK-FL is faster than FedAvg on the client side. In this section, we focus on the perspectives of the communication efficiency and security in terms of data confidentiality and membership privacy.

For communication, we follow the widely adopted analysis framework in wireless communication to examine only the client uplink overhead, assuming that the downlink bandwidth is much larger and the server will have enough transmission power (Tran et al., 2019). In NTK-FL, the client uplink communication cost and space complexity are dominated by a third-order tensor $\boldsymbol{J}_m^{(k)}$, i.e., an $O(N_m d_2 d)$ complexity compared to $O(d)$ in FedAvg. For security, we investigate a threat model where a curious server may perform membership inference attacks (Nasr et al., 2018) or data reconstruction attacks (Zhu et al., 2019). Compared to the averaged gradient, sample-wise Jacobian matrices are more expressive, which may facilitate such attacks from the aggregation server. We extend NTK-FL by combining various tools to solve the aforementioned problems without jeopardizing the performance severely. Although it is possible to incorporate these tools into FedAvg, we will show that overall it will lead to more severe accuracy drop.

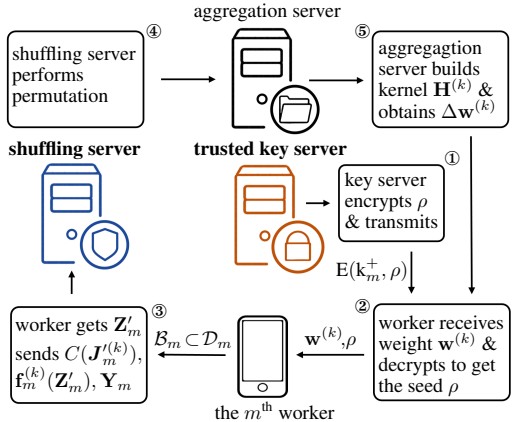

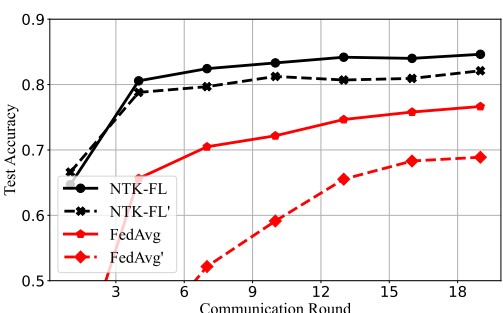

Figure 2: Schematic of CP-NTK-FL. A trusted key server (orange) sends an encrypted seed $E(k_m^+, \rho)$ with the public key $k_m^+$ for random projection. The client transmits the required message to the shuffling server (blue) for permutation.

Figure 3: Training results of 300 workers via NTK-FL and FedAvg, along with variants with the local dataset subsampling and random projection, denoted as NTK-FL$'$ and FedAvg$'$, respectively. We train a two-layer multilayer perceptron on the Fashion-MNIST dataset. The joint effect causes more accuracy degradation in FedAvg (red) than in NTK-FL (black).

**Jacobian Dimension Reduction.** First, we let the $m$th worker sample a subset $\mathcal{B}_m$ from its dataset $\mathcal{D}_m$ uniformly for the training. Let $\beta \in (0,1)$ denote the sampling rate, $\mathcal{B}_m$ contains $N'_m = \beta N_m$ data points, with the training pairs denoted by $(\mathbf{X}'_m, \mathbf{Y}'_m)$. Next, we consider using a random projection to preprocess the input data via a seed shared by a *trusted key server*. Formally, the sampled training examples are projected into $\mathbf{Z}'_m$, i.e., $\mathbf{Z}'_m = \mathbf{X}'_m \mathbf{P}$, where $\mathbf{P} \in \mathbb{R}^{d_1 \times d'_1}$ is a projection matrix generated based on a seed $\rho$ with IID standard Gaussian entries. In general, we have $d'_1 < d_1$ and an non-invertible projection operation. The concept of trusted key server follows the trusted third party in cryptography (Van Oorschot, 2020), and we assume it will not be compromised.

These two steps can already reduce the uplink communication overhead and enhance privacy. We first examine the current Jacobian tensor $\boldsymbol{J}'^{(k)}_m \in \mathbb{R}^{N'_m \times d_2 \times d'}$. Compared with its original version $\boldsymbol{J}^{(k)}_m$, it has reduced dimensionality at the cost of certain information loss. Meanwhile, the random projection will defend against the data reconstruction attack, as the Jacobian tensor is now evaluated at the projected data $\mathbf{Z}'_m$. We empirically verify their impact on the test accuracy in Figure 3. We set $d'_1 = 100$ and sampling rate $\beta = 0.4$, and train a multilayer perceptron with 100 hidden nodes on the Fashion-MNIST dataset (Xiao et al., 2017). The joint effect of these strategies is a slight accuracy drop in NTK-FL and a nonnegligible performance degradation in FedAvg.

**Jacobian Compression and Shuffling.** We use a compression scheme to reduce the size of the Jacobian tensor by zeroing out the coordinates with small magnitude (Alistarh et al., 2018). In addition to the communication efficiency, this compression scheme is empirically effective against the data reconstruction attack (Zhu et al., 2019). To further ensure the confidentiality and membership privacy, we introduce a *shuffling server*, inspired by some recent frameworks (Girgis et al., 2021; Cheng et al., 2021), to permute Jacobian tensors $\boldsymbol{J}^{(k)}_m$'s, neural network states $\mathbf{f}^{(k)}_m$'s, and labels $\mathbf{Y}_m$'s. Based on (7b), we denote the model update by $\Delta \mathbf{w}^{(k)} \triangleq \mathbf{w}^{(k+1)} - \mathbf{w}^{(k)} = \sum_{j=1}^{d_2} (\boldsymbol{J}^{(k)}_{:j:})^\top \mathbf{R}^{(k,t)}_{:j}$, which is a sum of matrix products. If rows and columns are permuted in synchronization, the weight update $\Delta \mathbf{w}^{(k)}$ will remain unchanged. Considering the high dimensionality of the neural network weight, the reconstruction attack becomes computationally infeasible. As provable differential privacy guarantee does not explicitly protect against the reconstruction attack (Zhang et al., 2020), we leave a thorough privacy study for future work.

## 5 ANALYSIS OF ALGORITHM

In this section, we analyze the loss decay rate between successive communication rounds in NTK-FL and make comparisons with FedAvg. Similar to Du et al. (2019) and Dukler et al. (2020), we

consider a two-layer neural network $f : \mathbb{R}^d \to \mathbb{R}$ of the following form to facilitate our analysis:

$$f(\mathbf{x}; \mathbf{V}, \mathbf{c}) = \frac{1}{\sqrt{n}} \sum_{r=1}^{n} c_r \sigma(\mathbf{v}_r^\top \mathbf{x}), \tag{10}$$

where $\mathbf{x} \in \mathbb{R}^{d_1}$ is an input, $\mathbf{v}_r \in \mathbb{R}^{d_1}$ is the weight vector in the first layer, $\mathbf{V} = [\mathbf{v}_1, \cdots, \mathbf{v}_n]$, $c_r \in \mathbb{R}$ is the weight in the second layer, and $\sigma(\cdot)$ is the rectified linear unit (ReLU) function, namely $\sigma(z) = \max(z, 0)$, applied coordinatewise. We state two assumptions as prerequisites.

**Assumption 1** *The first layer $\mathbf{v}_r$'s are sampled from $\mathcal{N}(0, \alpha^2 \mathbf{I})$. The second layer $c_r$'s are sampled from $\{-1, 1\}$ with equal probability and are kept fixed during training.*

Assumption 1 gives the initial distribution of the neural network parameters. Similar assumptions can be found in Dukler et al. (2020). We add restrictions to the input data in the next assumption.

**Assumption 2** *(Normalized input). The input data are normalized, i.e., $\|\mathbf{x}_i\|_2 \leqslant 1, \forall i$.*

For this neural network model, the $(i, j)$th entry of the empirical kernel matrix $\mathbf{H}^{(k)}$ given in (3.2) can be calculated as: $(\mathbf{H}^{(k)})_{ij} = \frac{1}{n} \mathbf{x}_i^\top \mathbf{x}_j \sum_{r=1}^{n} \mathbb{1}_{ir}^{(k)} \mathbb{1}_{jr}^{(k)}$, where $\mathbb{1}_{ir}^{(k)} \triangleq \mathbb{1}\{\langle \mathbf{v}_r^{(k)}, \mathbf{x}_i \rangle \geqslant 0\}$, and the term $c_r^2$ is omitted according to Assumption 1. Define $\mathbf{H}^\infty$, whose $(i, j)$th entry is given by:

$$(\mathbf{H}^\infty)_{ij} \triangleq \mathbb{E}_{\mathbf{v} \sim \mathcal{N}(0, \alpha^2 \mathbf{I})} \left[ \mathbf{x}_i^\top \mathbf{x}_j \mathbb{1}(\mathbf{v}^\top \mathbf{x}_i \geqslant 0) \mathbb{1}(\mathbf{v}^\top \mathbf{x}_j \geqslant 0) \right]. \tag{11}$$

Let $\lambda_0$ denote the minimum eigenvalue of $\mathbf{H}^\infty$, which is restricted in the next assumption.

**Assumption 3** *The kernel matrix $\mathbf{H}^\infty$ is positive definite, namely, $\lambda_0 > 0$.*

In fact, the positive-definite property of $\mathbf{H}^\infty$ can be shown under certain conditions (Dukler et al., 2020). For simplicity, we omit the proof details and directly assume the positive definiteness of $\mathbf{H}^\infty$ in Assumption 3. Next, we study the residual term $\|f^{(k)}(\mathbf{X}) - \mathbf{y}\|_2^2$ in communication round $k$, where $\mathbf{X} = [\mathbf{X}_1^\top, \ldots, \mathbf{X}_M^\top]^\top \in \mathbb{R}^{N \times d_1}$ denote a concatenation of client inputs and $\mathbf{y} = [\mathbf{y}_1^\top, \ldots, \mathbf{y}_M^\top]^\top \in \mathbb{R}^N$ denote a concatenation of client labels. We give the convergence result by analyzing how the residual term decays between successive rounds.

**Theorem 1** *For the NTK-FL scheme under Assumptions 1 to 3, let the learning rate $\eta = O\left(\frac{\lambda_0}{N}\right)$ and the neural network width $n = \Omega\left(\frac{N^2}{\lambda_0^2} \ln \frac{N^2}{\delta}\right)$, then with probability at least $1 - \delta$, the one-round loss decay of NTK-FL is*

$$\left\|f^{(k+1)}(\mathbf{X}) - \mathbf{y}\right\|_2^2 \leqslant \left(1 - \frac{\eta \lambda_0}{2N}\right)^{t^{(k)}} \left\|f^{(k)}(\mathbf{X}) - \mathbf{y}\right\|_2^2, \tag{12}$$

*where $t^{(k)}$ is the number of NTK update steps defined in (9).*

The proof of Theorem 1 can be found in Appendix B. By studying the asymmetric kernel matrix caused by local update (Huang et al., 2021), we have the following theorem for FedAvg, where the proof can be found in Appendix C.

**Theorem 2** *For FedAvg under Assumptions 1 to 3, let the learning rate $\eta = O\left(\frac{\lambda_0}{\tau N |\mathcal{C}_k|}\right)$ and the neural network width $n = \Omega\left(\frac{N^2}{\lambda_0^2} \ln \frac{N^2}{\delta}\right)$, then with probability at least $1 - \delta$, the one-round loss decay of FedAvg is*

$$\left\|f^{(k+1)}(\mathbf{X}) - \mathbf{y}\right\|_2^2 \leqslant \left(1 - \frac{\eta \tau \lambda_0}{2N |\mathcal{C}_k|}\right) \left\|f^{(k)}(\mathbf{X}) - \mathbf{y}\right\|_2^2, \tag{13}$$

*where $\tau$ is the number of local iterations, and $|\mathcal{C}_k|$ is the cardinality of the worker set in round $k$.*

**Remark 1** *(Fast Convergence of NTK-FL). The convergence rate of NTK-FL is faster than FedAvg. To see this, we compare the Binomial approximation of the decay coefficient in Theorem 1 with the decay coefficient in Theorem 2, i.e., $1 - \frac{\eta_1 t^{(k)} \lambda_0}{2N} + O\left(\eta_1^2\right) < 1 - \frac{\eta_2 \tau \lambda_0}{2N |\mathcal{C}_k|}$, where $\eta \ll 1$ for a large $N$ [1]. The number of NTK update steps $t^{(k)}$ is chosen dynamically in (9), which is on the order of $10^2$ to $10^3$, whereas $\tau$ is often on the order of magnitude of $10$ in literature (Reisizadeh et al., 2020; Haddadpour et al., 2021). One can verify that $\eta_1 t^{(k)} \lambda_0$ is larger than $\eta_2 \tau \lambda_0 / |\mathcal{C}_k|$ and draw the conclusion in (1).*

---

[1] For example, if we have 100 clients, each of which has more than 100 data points, then $N$ is on the order of $10^4$. Considering the choice of the learning rate $\eta_1$, the Binomial approximation holds in (1).

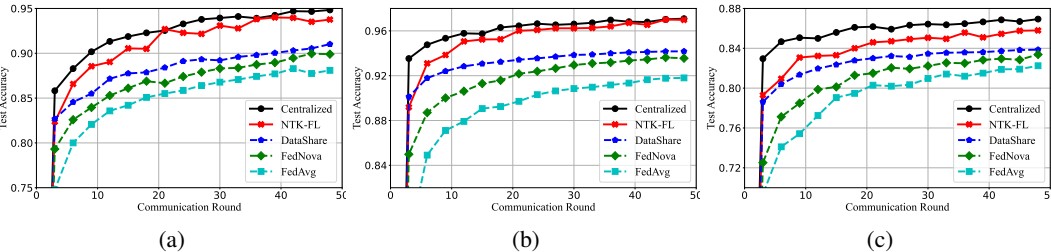

Figure 4: Test accuracy versus communication round of different methods evaluated on: (a) EM-NIST dataset, where the heterogeneity comes from feature skewness. (b) non-IID MNIST dataset with label skewness, where the Dirichlet distribution parameter $\alpha = 0.5$. (c) non-IID Fashion-MNIST dataset with label skewness, where the Dirichlet distribution parameter $\alpha = 0.5$. NTK-FL outperforms all baseline FL algorithms in different scenarios, and achieves similar test performance compared with the ideal centralized training case.

## 6 EXPERIMENTAL RESULTS

**Federated Settings.** We use three datasets, namely, MNIST (LeCun et al., 1998), Fashion-MNIST (Xiao et al., 2017), and EMNIST (Cohen et al., 2017) digits. All of them contain $C = 10$ categories. For MNIST and Fashion-MNIST, we follow Hsu et al. (2019) to simulate non-IID data with the symmetric Dirichlet distribution (Good, 1976). Specifically, for the $m$th worker, we draw a random vector $\boldsymbol{q}_m \sim \text{Dir}(\alpha)$, where $\boldsymbol{q}_m = [q_{m,1}, \ldots, q_{m,C}]^\top$ belongs to the $(C-1)$-standard simplex. Images with category $k$ are assigned to the $m$th worker in proportional to $(100 \cdot q_{m,k})\%$. The heterogeneity in this setting mainly comes from label skewness. For the EMNIST dataset, it has a federated version that splits the dataset into shards indexed by the original writer of the digits (Kairouz et al., 2021). The heterogeneity mainly comes from feature skewness. A multilayer perceptron model with 100 hidden nodes is chosen as the target neural network model. We consider a total of 300 workers and select 20 of them with equal probability in each round.

**Convergence.** We empirically verify the convergence rate of the proposed method. For FedAvg, we use the number of local iterations from $\{10, 20, \ldots, 50\}$ and report the best results. For NTK-FL, we choose $t^{(k)}$ over the set $\{100, 200, \ldots, 2000\}$. We use the following methods that are robust to the non-IID setting as the baselines: (i) Data sharing scheme suggested by Zhao et al. (2018), where a global dataset is broadcasted to workers for local training; the size of the global dataset is set to be $10\%$ of the total number of local data points. (ii) Federated normalized averaging (FedNova) (Wang et al., 2020), where the workers transmit normalized gradient vectors to the server. (iii) Centralized training simulation, where the server collects the data points from subset $\mathcal{C}_k$ of workers and performs gradient descent to directly train the global model. Clearly, scheme (iii) achieves the performance that can be considered as an upper bound of all other algorithms. The training curves over three repetitions are shown in Figure 4. More implementation details and the results on CIFAR-10 (Krizhevsky, 2009) can be found in Appendix A. Our proposed NTK-FL method shows consistent advantages over other methods in different non-IID scenarios.

**Degree of Heterogeneity.** In this experiment, we select the Dirichlet distribution parameter $\alpha$ from $\{0.1, 0.2, 0.3, 0.4, 0.5\}$ and simulate different degrees of heterogeneity on Fashion-MNIST dataset. A smaller $\alpha$ will increase the degree of heterogeneity in the data distribution. We evaluate NTK-FL, DataShare, FedNova, and FedAvg model test accuracy after training for 50 rounds. The mean values over three repetitions are shown in Figure 5, where each point is obtained over five repetitions with standard deviation less than $1\%$. It can be observed that NTK-FL achieves stable test accuracy in different heterogeneous settings. In comparison, FedAvg and FedNova show a performance drop in the small $\alpha$ region. NTK-FL has more advantages over baselines methods when the degree of heterogeneity is larger.

**Effect of Hyperparameters.** We study the effect of the tunable parameters in CP-NTK-FL. We change the local data sampling rate $\beta$ and dimension $d'_1$, and evaluate the model test accuracy on the non-IID Fashion-MNIST dataset ($\alpha = 0.1$) after 10 communication rounds. The results are shown in Figure 6. A larger data sampling rate $\beta$ or a larger dimension $d'_1$ will cause less information loss,

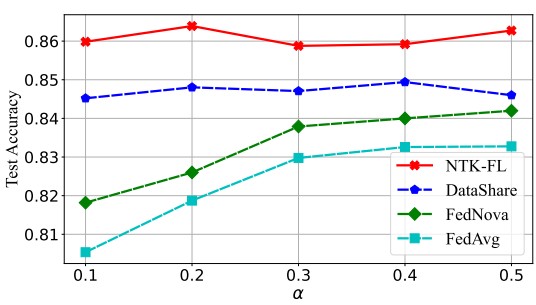

Figure 5: Test accuracy versus the Dirichlet distribution parameter $\alpha$ for different methods evaluated on the non-IID Fashion-MNIST dataset. Reducing the value of $\alpha$ will increase the degree of heterogeneity in the data distribution. NTK-FL is robust to different heterogeneous data distributions, and shows more advantages over FedAvg and FedNova when the degree of heterogeneity is larger.

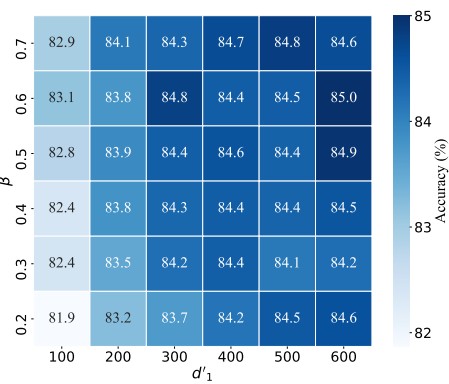

Figure 6: CP-NTK-FL test accuracy for different hyperparams. A larger data sampling rate $\beta$ and a larger dimension $d'_1$ are expected to give a higher test accuracy. In general, the scheme is robust to different combinations of hyperparameters.

and are expected to achieve a higher test accuracy. The results also show that the scheme is robust to different combinations of hyperparameters.

**Uplink Communication.** We evaluate the uplink communication efficiency of CP-NTK-FL ($d'_1 = 200, \beta = 0.3$) by measuring the number of rounds and cumulative uplink communication cost to reach a test accuracy of $85\%$ on non-IID Fashion-MNIST dataset ($\alpha = 0.1$). The results over three repetitions are shown in Table 1. Compared with federated learning with compression (FedCOM) (Haddadpour et al., 2021), quantized SGD (QSGD) (Alistarh et al., 2017), and FedAvg, CP-NTK-FL achieves the goal within an order of magnitude fewer iterations, which is particularly advantageous for applications with nonnegligible encoding/decoding delays or network latency.

Table 1: Uplink communication cost to reach $85\%$ on non-IID Fashion-MNIST dataset ($\alpha = 0.1$). CP-NTK-FL can achieve the target goal within the fewest communication rounds without incurring communication cost significantly.

| optimization algorithms | comm. rounds | comm. cost (MB) |
| --- | --- | --- |
| CP-NTK-FL | 26 | 386 |
| FedCOM | 250 | 379 |
| QSGD (4 bit) | 614 | 465 |
| FedAvg | 284 | 1720 |

## 7 CONCLUSION AND FUTURE WORK

In this paper, we have proposed an NTK empowered FL paradigm. It inherently solves the statistical heterogeneity challenge. By constructing a global kernel based on the local sample-wise Jacobian matrices, the global model weights can be updated via NTK evolution in the parameter space. Compared with traditional algorithms such as FedAvg, NTK-FL has a more centralized training flavor by transmitting more expressive updates. The effectiveness of the proposed paradigm has been verified theoretically and experimentally.

In future work, it will be interesting to extend the paradigm for other neural network architectures, such as CNNs, residual networks (ResNets) (He et al., 2016), and RNNs. It is also worthwhile to further improve the efficiency of NTK-FL and explore its savings in wall-clock time. We believe the proposed paradigm will provide a new perspective to solve federated learning challenges.

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

# A ADDITIONAL EXPERIMENTAL RESULTS

We give the detailed setting of the learning rate and batch size. For the learning rate $\eta$, we search over the set $\{10^{-3}, 3 \times 10^{-3}, 10^{-2}, 3 \times 10^{-2}, 10^{-1}\}$. The learning rate is fixed during the training. For the client batch size, we set it to 200 for all datasets. We evaluate different methods, including the centralized training simulation, data sharing method (Zhao et al., 2018), FedNova (Wang et al., 2020), FedAvg (McMahan et al., 2017), and the proposed NTK-FL on the non-IID CIFAR-10 dataset (Krizhevsky, 2009) and present the results in Figure 7. NTK-FL outperforms other FL algorithms and shows test accuracy close to the centralized simulation. The observation is consistent with the results in Figure 4.

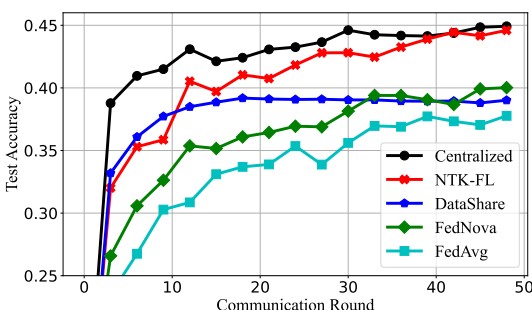

Figure 7: Test accuracy versus communication round of different methods evaluated on the non-IID CIFAR-10 dataset, where the Dirichlet distribution parameter $\alpha = 0.1$.

# B PROOF OF THEOREM 1

Let $\mathcal{I}_m$ denote a set of indices such that for $i \in \mathcal{I}_m$, $(\mathbf{x}_i, y_i) \in \mathcal{D}_m$. We first present some lemmas to facilitate the convergence analysis. In communication round $k$, define $\mathcal{S}_i^{(k)}$ as the set of indices corresponding to neurons whose activation pattern is similar to its initial state for an input $\mathbf{x}_i$:

$$\mathcal{S}_i^{(k)} \triangleq \left\{ r \in \{1, \ldots, n\} \big| \exists\, \mathbf{v}, \|\mathbf{v} - \mathbf{v}_r^{(0)}\|_2 \leqslant R, \mathbb{1}_{ir}^{(0)} \neq \mathbb{1}\left(\mathbf{v}^\top \mathbf{x}_i \geqslant 0\right) \right\}. \tag{14}$$

We upper bound the cardinality of $\mathcal{S}_i$ in Lemma 1.

**Lemma 1** *Under Assumption 1 to 2, with probability at least $1 - \delta$, we have*

$$|\mathcal{S}_i| \leqslant \sqrt{\frac{2}{\pi}} \frac{nR}{\delta\alpha}, \quad \forall\, i \in \{1, \ldots, N\}. \tag{15}$$

*Proof.* To bound $|\mathcal{S}_i| = \sum_{r=1}^{n} \mathbb{1}\left(r \in \mathcal{S}_i\right)$, consider an event $A_{ir}$ defined as follows:

$$A_{ir} \triangleq \{\exists\, \mathbf{v}, \|\mathbf{v} - \mathbf{v}_r^{(0)}\|_2 \leqslant R, \mathbb{1}_{ir}^{(0)} \neq \mathbb{1}\left(\mathbf{v}^\top \mathbf{x}_i \geqslant 0\right)\}. \tag{16}$$

Clearly, $\mathbb{1}\left(r \in \mathcal{S}_i\right) = \mathbb{1}\left(A_{ir}\right)$. According to Assumption 2, $\|\mathbf{x}_i\| \leqslant 1$, it can be shown that the event $A_{ir}$ happens if and only if $|(\mathbf{v}_r^{(0)})^\top \mathbf{x}_i| \leqslant R$ based on a geometric argument. Based on Assumption 1, we have $(\mathbf{v}_r^{(0)})^\top \mathbf{x}_i \sim \mathcal{N}(0, \alpha^2)$. The probability of event $A_{ir}$ is

$$\mathbb{P}[A_{ir}] = \mathbb{P}\left[|(\mathbf{v}_r^{(0)})^\top \mathbf{x}_i| \leqslant R\right] \tag{17a}$$

$$= \mathrm{erf}\left(\frac{R}{\sqrt{2}\alpha}\right) \leqslant \sqrt{\frac{2}{\pi}} \frac{R}{\alpha}. \tag{17b}$$

By Markov's inequality, we have with probability at least $1 - \delta$,

$$\sum_{r=1}^{n} \mathbb{1}\left(r \in \mathcal{S}_i\right) \leqslant \sqrt{\frac{2}{\pi}} \frac{nR}{\delta\alpha}. \tag{18}$$

The proof is complete. $\qquad\square$

We bound the perturbation of the kernel matrix $\mathbf{H}^{(k,t)}$ in Lemma 2.

**Lemma 2** *Under Assumption 1 to 2, if $\forall\, r \in \{1, \ldots, n\}$, $\|\mathbf{v}_r^{(k,t)} - \mathbf{v}_r^{(0)}\|_2 \leqslant R$, then*

$$\|\mathbf{H}^{(k,t)} - \mathbf{H}^{(0)}\|_2 \leqslant \frac{2\sqrt{2}NR}{\sqrt{\pi}\delta\alpha}. \tag{19}$$

*Proof.* We have

$$\|\mathbf{H}^{(k,t)} - \mathbf{H}^{(0)}\|_2^2 \leqslant \|\mathbf{H}^{(k,t)} - \mathbf{H}^{(0)}\|_{\mathrm{F}}^2 \tag{20a}$$

$$= \sum_{i=1}^{N} \sum_{j=1}^{N} \left[ (\mathbf{H}^{(k,t)})_{ij} - (\mathbf{H}^{(0)})_{ij} \right]^2 \tag{20b}$$

$$= \frac{1}{n^2} \sum_{i=1}^{N} \sum_{j=1}^{N} (\mathbf{x}_i^\top \mathbf{x}_j)^2 \left( \sum_{r=1}^{n} \mathbb{1}_{ir}^{(k,t)} \mathbb{1}_{jr}^{(k,t)} - \mathbb{1}_{ir}^{(0)} \mathbb{1}_{jr}^{(0)} \right)^2 . \tag{20c}$$

Consider the event $A_{ir}$ defined in (16). Let $\phi_{ijr}^{(k,t)} \triangleq \mathbb{1}_{ir}^{(k,t)} \mathbb{1}_{jr}^{(k,t)} - \mathbb{1}_{ir}^{(0)} \mathbb{1}_{jr}^{(0)}$. If $\neg A_{ir}$ and $\neg A_{jr}$ happen, clearly we have $|\phi_{ijr}^{(k,t)}| = 0$. Therefore, the expectation of $|\phi_{ijr}^{(k,t)}|$ can be bounded as

$$\mathbb{E}\left[ \left| \phi_{ijr}^{(k,t)} \right| \right] \leqslant \mathbb{P}(A_{ir} \cup A_{jr}) \cdot 1 \tag{21a}$$

$$\leqslant \mathbb{P}(A_{ir}) + \mathbb{P}(A_{jr}) \tag{21b}$$

$$\overset{\textcircled{1}}{\leqslant} 2\sqrt{\frac{2}{\pi}} \frac{R}{\alpha}, \tag{21c}$$

where $\textcircled{1}$ comes from (17b). By Markov's inequality, we have with probability at least $1 - \delta$,

$$|\phi_{ijr}^{(k,t)}| \leqslant 2\sqrt{\frac{2}{\pi}} \frac{R}{\delta\alpha}. \tag{22}$$

Plugging (22) into (20c) yields

$$\|\mathbf{H}^{(k,t)} - \mathbf{H}^{(0)}\|_2^2 \leqslant \frac{N^2}{n^2} \frac{8n^2 R^2}{\pi\delta^2\alpha^2} = \frac{8N^2 R^2}{\pi\delta^2\alpha^2}. \tag{23}$$

Taking the square root on both sides completes the proof. $\square$

**Lemma 3** *With probability at least $1 - \delta$,*

$$\|\mathbf{H}^{(0)} - \mathbf{H}^{\infty}\|_2 \leqslant N\sqrt{\frac{\ln\left(2N^2/\delta\right)}{2n}}. \tag{24}$$

*Proof.* We have

$$\|\mathbf{H}^{(0)} - \mathbf{H}^{\infty}\|_2^2 \leqslant \|\mathbf{H}^{(0)} - \mathbf{H}^{\infty}\|_{\mathrm{F}}^2 = \sum_{i=1}^{N} \sum_{j=1}^{N} \left[ (\mathbf{H}^{(0)})_{ij} - (\mathbf{H}^{\infty})_{ij} \right]^2 . \tag{25}$$

Note that $(\mathbf{H}^{(0)})_{ij} = \frac{1}{n} \mathbf{x}_i^\top \mathbf{x}_j \sum_{r=1}^{n} \mathbb{1}_{ir}^{(0)} \mathbb{1}_{jr}^{(0)}$, $(\mathbf{H}^{(0)})_{ij} \in [-1, 1]$. By Hoeffding's inequality, we have with probability at least $1 - \delta/n^2$,

$$\left| (\mathbf{H}^{(0)})_{ij} - (\mathbf{H}^{\infty})_{ij} \right| \leqslant \sqrt{\frac{\ln\left(2N^2/\delta\right)}{2n}}. \tag{26}$$

Applying the union bound over $i, j \in [N]$ yields

$$\|\mathbf{H}^{(0)} - \mathbf{H}^{\infty}\|_2 \leqslant N\sqrt{\frac{\ln\left(2N^2/\delta\right)}{2n}}. \tag{27}$$

The proof is complete. $\square$

Now we are going to prove Theorem 1.

**Theorem 1** *For the NTK-FL scheme under Assumptions 1 to 3, let the learning rate $\eta = O\left(\frac{\lambda_0}{N}\right)$ and the neural network width $n = \Omega\left(\frac{N^2}{\lambda_0^2} \ln \frac{2N^2}{\delta}\right)$, then with probability at least $1 - \delta$, the one-round loss decay of NTK-FL is*

$$\|f^{(k+1)}(\mathbf{X}) - \mathbf{y}\|_2^2 \leq \left( 1 - \frac{\eta\lambda_0}{2N} \right)^{t^{(k)}} \|f^{(k)}(\mathbf{X}) - \mathbf{y}\|_2^2. \tag{28}$$

*Proof.* Taking the difference between successive terms yields

$$f^{(k,t+1)}(\mathbf{x}_i) - f^{(k,t)}(\mathbf{x}_i) = \frac{1}{\sqrt{n}} \sum_{r=1}^{n} \left[ c_r \sigma \left( (\mathbf{v}_r^{(k,t+1)})^\top \mathbf{x}_i \right) - c_r \sigma \left( (\mathbf{v}_r^{(k,t)})^\top \mathbf{x}_i \right) \right]. \qquad (29)$$

We decompose the difference term to the sum of $d_i^{\mathrm{I}}$ and $d_i^{\mathrm{II}}$, based on the set $\mathcal{S}_i$:

$$d_i^{\mathrm{I}} \triangleq \frac{1}{\sqrt{n}} \sum_{r \notin \mathcal{S}_i} \left[ c_r \sigma \left( (\mathbf{v}_r^{(k,t+1)})^\top \mathbf{x}_i \right) - c_r \sigma \left( (\mathbf{v}_r^{(k,t)})^\top \mathbf{x}_i \right) \right], \qquad (30a)$$

$$d_i^{\mathrm{II}} \triangleq \frac{1}{\sqrt{n}} \sum_{r \in \mathcal{S}_i} \left[ c_r \sigma \left( (\mathbf{v}_r^{(k,t+1)})^\top \mathbf{x}_i \right) - c_r \sigma \left( (\mathbf{v}_r^{(k,t)})^\top \mathbf{x}_i \right) \right]. \qquad (30b)$$

Consider the residual term

$$\left\| f^{(k,t+1)}(\mathbf{X}) - \mathbf{y} \right\|_2^2 \qquad (31a)$$

$$= \left\| f^{(k,t+1)}(\mathbf{X}) - f^{(k,t)}(\mathbf{X}) + f^{(k,t)}(\mathbf{X}) - \mathbf{y} \right\|_2^2 \qquad (31b)$$

$$= \left\| f^{(k,t)}(\mathbf{X}) - \mathbf{y} \right\|_2^2 + 2 \left\langle \mathbf{d}^{\mathrm{I}} + \mathbf{d}^{\mathrm{II}}, f^{(k,t)}(\mathbf{X}) - \mathbf{y} \right\rangle + \left\| f^{(k,t+1)}(\mathbf{X}) - f^{(k,t)}(\mathbf{X}) \right\|_2^2. \qquad (31c)$$

We will give upper bounds for the inner product terms $\left\langle \mathbf{d}^{\mathrm{I}}, f^{(k,t)}(\mathbf{X}) - \mathbf{y} \right\rangle$, $\left\langle \mathbf{d}^{\mathrm{II}}, f^{(k,t)}(\mathbf{X}) - \mathbf{y} \right\rangle$, and the difference term $\left\| f^{(k,t+1)}(\mathbf{X}) - f^{(k,t)}(\mathbf{X}) \right\|_2^2$, separately. Based on the property of the set $\mathcal{S}_i$, we have

$$d_i^{\mathrm{I}} = -\frac{\eta}{\sqrt{n}} \sum_{r \notin \mathcal{S}_i} c_r \left\langle \nabla_{\mathbf{v}_r} L, \mathbf{x}_i \right\rangle \mathbb{1}_{ir}^{(k,t)} \qquad (32a)$$

$$= -\frac{\eta}{nN} \sum_{j=1}^{N} \left( f^{(k,t)}(\mathbf{x}_j) - y_j \right) \mathbf{x}_j^\top \mathbf{x}_i \sum_{r \notin \mathcal{S}_i} c_r^2 \, \mathbb{1}_{ir}^{(k,t)} \mathbb{1}_{jr}^{(k,t)} \qquad (32b)$$

$$= -\frac{\eta}{N} \sum_{j=1}^{N} \left( f^{(k,t)}(\mathbf{x}_j) - y_j \right) \left( (\mathbf{H}^{(k,t)})_{ij} - (\mathbf{H}^{\perp(k,t)})_{ij} \right), \qquad (32c)$$

where $(\mathbf{H}^{\perp(k,t)})_{ij}$ is defined as

$$(\mathbf{H}^{\perp(k,t)})_{ij} \triangleq \frac{1}{n} \mathbf{x}_i^\top \mathbf{x}_j \sum_{r \in \mathcal{S}_i}^{n} \mathbb{1}_{ir}^{(k,t)} \mathbb{1}_{jr}^{(k,t)}. \qquad (33)$$

For the inner product term $\left\langle \mathbf{d}^{\mathrm{I}}, f^{(k,t)}(\mathbf{X}) - \mathbf{y} \right\rangle$, we have

$$\left\langle \mathbf{d}^{\mathrm{I}}, f^{(k,t)}(\mathbf{X}) - \mathbf{y} \right\rangle = -\frac{\eta}{N} (f^{(k,t)}(\mathbf{X}) - \mathbf{y})^\top (\mathbf{H}^{(k,t)} - \mathbf{H}^{\perp(k,t)})(f^{(k,t)}(\mathbf{X}) - \mathbf{y}). \qquad (34)$$

Let $T_1$ and $T_2$ denote the following terms

$$T_1 \triangleq -(f^{(k,t)}(\mathbf{X}) - \mathbf{y})^\top \mathbf{H}^{(k,t)}(f^{(k,t)}(\mathbf{X}) - \mathbf{y}), \qquad (35a)$$

$$T_2 \triangleq (f^{(k,t)}(\mathbf{X}) - \mathbf{y})^\top \mathbf{H}^{\perp(k,t)}(f^{(k,t)}(\mathbf{X}) - \mathbf{y}). \qquad (35b)$$

With probability at least $1 - \delta$, $T_1$ can be bounded as:

$$T_1 = -(f^{(k,t)}(\mathbf{X}) - \mathbf{y})^\top (\mathbf{H}^{(k,t)} - \mathbf{H}^{(0)} + \mathbf{H}^{(0)} - \mathbf{H}^\infty + \mathbf{H}^\infty)(f^{(k,t)}(\mathbf{X}) - \mathbf{y}) \qquad (36a)$$

$$\leqslant -(f^{(k,t)}(\mathbf{X}) - \mathbf{y})^\top (\mathbf{H}^{(k,t)} - \mathbf{H}^{(0)})(f^{(k,t)}(\mathbf{X}) - \mathbf{y})$$
$$\quad - (f^{(k,t)}(\mathbf{X}) - \mathbf{y})^\top (\mathbf{H}^{(0)} - \mathbf{H}^\infty)(f^{(k,t)}(\mathbf{X}) - \mathbf{y}) - \lambda_0 \left\| f^{(k,t)}(\mathbf{X}) - \mathbf{y} \right\|_2^2 \qquad (36b)$$

$$\overset{\text{①}}{\leqslant} \left( \frac{2\sqrt{2}NR}{\sqrt{\pi}\delta\alpha} + N\sqrt{\frac{\ln(2N^2/\delta)}{2n}} - \lambda_0 \right) \left\| f^{(k,t)}(\mathbf{X}) - \mathbf{y} \right\|_2^2, \qquad (36c)$$

where ① comes from Lemma 2 and Lemma 3. To bound the term $T_2$, consider the $\ell_2$ norm of the matrix $\mathbf{H}^{\perp(k,t)}$. With probability at least $1 - \delta$, we have:

$$\|\mathbf{H}^{\perp(k,t)}\|_2 \leqslant \|\mathbf{H}^{\perp(k,t)}\|_{\mathrm{F}} \tag{37a}$$

$$= \left( \sum_{i=1}^{N} \sum_{j=1}^{N} \left( \frac{1}{n} \sum_{r \in \mathcal{S}_i} \mathbf{x}_i^\top \mathbf{x}_j \mathbb{1}_{ir}^{(k,t)} \mathbb{1}_{jr}^{(k,t)} \right)^2 \right)^{\frac{1}{2}} \tag{37b}$$

$$\leqslant \frac{N}{n} |\mathcal{S}_i| \overset{①}{\leqslant} \sqrt{\frac{2}{\pi}} \frac{NR}{\delta \alpha}, \tag{37c}$$

where ① comes from Lemma 1. Therefore, with probability at least $1 - \delta$, we have

$$T_2 \leqslant \sqrt{\frac{2}{\pi}} \frac{NR}{\delta \alpha} \| f^{(k,t)}(\mathbf{X}) - \mathbf{y} \|_2^2. \tag{38}$$

Combine the results of (36c) and (38):

$$\left\langle \mathbf{d}^{\mathrm{I}}, f^{(k,t)}(\mathbf{X}) - \mathbf{y} \right\rangle \leqslant \eta \left( \frac{3\sqrt{2}R}{\sqrt{\pi}\delta \alpha} + \sqrt{\frac{\ln(2N^2/\delta)}{2n}} - \frac{\lambda_0}{N} \right) \| f^{(k,t)}(\mathbf{X}) - \mathbf{y} \|_2^2. \tag{39}$$

For the inner product term $\left\langle \mathbf{d}^{\mathrm{II}}, f^{(k,t)}(\mathbf{X}) - \mathbf{y} \right\rangle$, we first bound $\|\mathbf{d}^{\mathrm{II}}\|_2^2$ as follows:

$$\|\mathbf{d}^{\mathrm{II}}\|_2^2 = \sum_{i=1}^{N} \left( \frac{1}{\sqrt{n}} \sum_{r \in \mathcal{S}_i} \left[ c_r \sigma \left( (\mathbf{v}_r^{(k,t+1)})^\top \mathbf{x}_i \right) - c_r \sigma \left( (\mathbf{v}_r^{(k,t)})^\top \mathbf{x}_i \right) \right] \right)^2 \tag{40a}$$

$$\overset{①}{\leqslant} \frac{\eta^2}{n} \sum_{i=1}^{N} |\mathcal{S}_i| \sum_{r \in \mathcal{S}_i} (c_r \langle \nabla_{\mathbf{v}_r} L, \mathbf{x}_i \rangle)^2 \tag{40b}$$

$$\overset{②}{\leqslant} \frac{\eta^2}{n} \sum_{i=1}^{N} |\mathcal{S}_i| \sum_{r \in \mathcal{S}_i} \|\nabla_{\mathbf{v}_r} L\|_2^2 \|\mathbf{x}_i\|_2^2 \tag{40c}$$

$$\leqslant \frac{\eta^2 N}{n} |\mathcal{S}_i|^2 \max_{r \in [n]} \|\nabla_{\mathbf{v}_r} L\|_2^2 \tag{40d}$$

$$\leqslant \frac{\eta^2 |\mathcal{S}_i|^2}{n^2} \| f^{(k,t)}(\mathbf{X}) - \mathbf{y} \|_2^2, \tag{40e}$$

where ① comes from the Lipschitz continuity of the ReLU function $\sigma(\cdot)$, ② holds due to Cauchy–Schwartz inequality. Plug (18) into (40e), we have with probability at least $1 - \delta$:

$$\|\mathbf{d}^{\mathrm{II}}\|_2^2 \leqslant \frac{2\eta^2 R^2}{\pi \delta^2 \alpha^2} \| f^{(k,t)}(\mathbf{X}) - \mathbf{y} \|_2^2. \tag{41}$$

The inner product term $\left\langle \mathbf{d}^{\mathrm{II}}, f^{(k,t)}(\mathbf{X}) - \mathbf{y} \right\rangle$ can be bounded as

$$\left\langle \mathbf{d}^{\mathrm{II}}, f^{(k,t)}(\mathbf{X}) - \mathbf{y} \right\rangle \leqslant \frac{\sqrt{2}\eta R}{\sqrt{\pi}\delta \alpha} \| f^{(k,t)}(\mathbf{X}) - \mathbf{y} \|_2^2. \tag{42}$$

Finally, the bound for the difference term is derived as

$$\left\| f^{(k,t+1)}(\mathbf{X}) - f^{(k,t)}(\mathbf{X}) \right\|_2^2 \leqslant \sum_{i=1}^{N} \left( \frac{\eta}{\sqrt{n}} \sum_{r=1}^{n} c_r \langle \nabla_{\mathbf{v}_r} L, \mathbf{x}_i \rangle \right)^2 \leqslant \eta^2 \| f^{(k,t)}(\mathbf{X}) - \mathbf{y} \|_2^2. \tag{43}$$

Combine the results of (39), (42) and (43):

$$\left\| f^{(k,t+1)}(\mathbf{X}) - \mathbf{y} \right\|_2^2 \leqslant \left[ 1 + \frac{8\sqrt{2}\eta R}{\sqrt{\pi}\delta \alpha} + 2\eta \sqrt{\frac{\ln(2N^2/\delta)}{2n}} - \frac{2\eta \lambda_0}{N} + \eta^2 \right] \| f^{(k,t)}(\mathbf{X}) - \mathbf{y} \|_2^2. \tag{44}$$

Let $R = O\left( \frac{\delta \alpha \lambda_0}{N} \right)$, $n = \Omega\left( \frac{N^2}{\lambda_0^2} \ln \frac{N^2}{\delta} \right)$, and $\eta = O\left( \frac{\lambda_0}{N} \right)$, we have

$$\left\| f^{(k,t+1)}(\mathbf{X}) - \mathbf{y} \right\|_2^2 \leqslant \left( 1 - \frac{\eta \lambda_0}{2N} \right) \| f^{(k,t)}(\mathbf{X}) - \mathbf{y} \|_2^2. \tag{45}$$

Summing up over the selected number $t^{(k)}$ iterations completes the proof. $\qquad \square$

## C    PROOF OF THEOREM 2

**Theorem 2** *For FedAvg under Assumptions 1 to 3, let the learning rate $\eta = O\left(\frac{\lambda_0}{\tau N |\mathcal{C}_k|}\right)$ and the neural network width $n = \Omega\left(\frac{N^2}{\lambda_0^2} \ln \frac{2N^2}{\delta}\right)$, then with probability at least $1 - \delta$, the one-round loss decay of FedAvg is*

$$\left\| f^{(k+1)}(\mathbf{X}) - \mathbf{y} \right\|_2^2 \leqslant \left(1 - \frac{\eta \tau \lambda_0}{2N |\mathcal{C}_k|}\right) \left\| f^{(k)}(\mathbf{X}) - \mathbf{y} \right\|_2^2. \tag{46}$$

*Proof.* We first construct a different set of kernel matrices $\{\mathbf{\Lambda}^{(k)}, \mathbf{\Lambda}_m^{(k,\tau)}\}$ similar to Huang et al. (2021). Let $\mathbb{1}_{imr}^{(k,u)} \triangleq \mathbb{1}\{\langle \mathbf{v}_{m,r}^{(k,u)}, \mathbf{x}_i \rangle \geqslant 0\}$, the $(i,j)$th entry of $\mathbf{\Lambda}_m^{(k,u)}$ and $\mathbf{\Lambda}^{(k,u)}$ is defined as

$$(\mathbf{\Lambda}_m^{(k,u)})_{ij} \triangleq \frac{1}{n} \mathbf{x}_i^\top \mathbf{x}_j \sum_{r=1}^n \mathbb{1}_{imr}^{(k,0)} \mathbb{1}_{jmr}^{(k,u)}, \tag{47a}$$

$$(\mathbf{\Lambda}^{(k,u)})_{ij} \triangleq (\mathbf{\Lambda}_m^{(k,u)})_{ij}, \quad \text{if } (\mathbf{x}_j, y_j) \in \mathcal{D}_m. \tag{47b}$$

Taking the difference between successive terms yields

$$f^{(k+1)}(\mathbf{x}_i) - f^{(k)}(\mathbf{x}_i) = \frac{1}{\sqrt{n}} \sum_{r=1}^n \left[ c_r \sigma\left((\mathbf{v}_r^{(k+1)})^\top \mathbf{x}_i\right) - c_r \sigma\left((\mathbf{v}_r^{(k)})^\top \mathbf{x}_i\right) \right]. \tag{48}$$

We decompose the difference term to the sum of $d_i^{\mathrm{I}}$ and $d_i^{\mathrm{II}}$, based on the set $\mathcal{S}_i$ and its complement:

$$d_i^{\mathrm{I}} \triangleq \frac{1}{\sqrt{n}} \sum_{r \notin \mathcal{S}_i} \left[ c_r \sigma\left((\mathbf{v}_r^{(k+1)})^\top \mathbf{x}_i\right) - c_r \sigma\left((\mathbf{v}_r^{(k)})^\top \mathbf{x}_i\right) \right], \tag{49a}$$

$$d_i^{\mathrm{II}} \triangleq \frac{1}{\sqrt{n}} \sum_{r \in \mathcal{S}_i} \left[ c_r \sigma\left((\mathbf{v}_r^{(k+1)})^\top \mathbf{x}_i\right) - c_r \sigma\left((\mathbf{v}_r^{(k)})^\top \mathbf{x}_i\right) \right]. \tag{49b}$$

Consider the residual term

$$\left\| f^{(k+1)}(\mathbf{X}) - \mathbf{y} \right\|_2^2 \tag{50a}$$

$$= \left\| f^{(k+1)}(\mathbf{X}) - f^{(k)}(\mathbf{X}) + f^{(k)}(\mathbf{X}) - \mathbf{y} \right\|_2^2 \tag{50b}$$

$$= \left\| f^{(k)}(\mathbf{X}) - \mathbf{y} \right\|_2^2 + 2 \left\langle \mathbf{d}^{\mathrm{I}} + \mathbf{d}^{\mathrm{II}}, f^{(k)}(\mathbf{X}) - \mathbf{y} \right\rangle + \left\| f^{(k+1)}(\mathbf{X}) - f^{(k)}(\mathbf{X}) \right\|_2^2. \tag{50c}$$

We will give upper bounds for the inner product terms $\left\langle \mathbf{d}^{\mathrm{I}}, f^{(k)}(\mathbf{X}) - \mathbf{y} \right\rangle$, $\left\langle \mathbf{d}^{\mathrm{II}}, f^{(k)}(\mathbf{X}) - \mathbf{y} \right\rangle$, and the difference term $\left\| f^{(k+1)}(\mathbf{X}) - f^{(k)}(\mathbf{X}) \right\|_2^2$, separately. For an input $\mathbf{x} \in \mathbb{R}^{d_1}$, let $f_m^{(k,u)}(\mathbf{x}) \triangleq \frac{1}{\sqrt{n}} \sum_{r=1}^n c_r \sigma(\langle \mathbf{v}_{m,r}^{(k,u)} \rangle, \mathbf{x}))$. By the update rule of FedAvg, the relation between the weight vector $\mathbf{v}_r^{(k)}$ in successive communication rounds is:

$$\mathbf{v}_r^{(k+1)} = \mathbf{v}_r^{(k)} - \frac{\eta}{|\mathcal{C}_k|} \sum_{m \in \mathcal{C}_k} \sum_{u=0}^{\tau-1} \nabla L_{\mathbf{v}_r^{(k,u)}} \tag{51a}$$

$$= \mathbf{v}_r^{(k)} - \frac{\eta c_r}{N \sqrt{n} |\mathcal{C}_k|} \sum_{m \in \mathcal{C}_k} \sum_{u=0}^{\tau-1} \sum_{j \in \mathcal{I}_m} (f_m^{(k,u)}(\mathbf{x}_j) - y_j) \mathbf{x}_j \mathbb{1}_{jmr}^{(k,u)}. \tag{51b}$$

Based on the property of the set $\mathcal{S}_i$, we have

$$d_i^{\mathrm{I}} = -\frac{1}{\sqrt{n}} \sum_{m \in \mathcal{C}_k} \sum_{u=0}^{\tau-1} \sum_{r \notin S_i} c_r \left\langle \mathbf{v}_r^{(k+1)} - \mathbf{v}_r^{(k)}, \mathbf{x}_i \right\rangle \mathbb{1}_{ir}^{(k)} \tag{52a}$$

$$= -\frac{\eta}{Nn |\mathcal{C}_k|} \sum_{m \in \mathcal{C}_k} \sum_{u=0}^{\tau-1} \sum_{r \notin S_i} \sum_{j \in \mathcal{I}_m} (f_m^{(k,u)}(\mathbf{x}_j) - y_j) \mathbf{x}_i^\top \mathbf{x}_j \mathbb{1}_{ir}^{(k)} \mathbb{1}_{jmr}^{(k,u)} \tag{52b}$$

$$= -\frac{\eta}{N |\mathcal{C}_k|} \sum_{m \in \mathcal{C}_k} \sum_{u=0}^{\tau-1} \sum_{j \in \mathcal{I}_m} (f_m^{(k,u)}(\mathbf{x}_j) - y_j) \left[ (\mathbf{\Lambda}_m^{(k,u)})_{ij} - (\mathbf{\Lambda}_m^{\perp(k,u)})_{ij} \right]. \tag{52c}$$

For the inner product term $\langle \mathbf{d}^{\mathrm{I}}, f^{(k)}(\mathbf{X}) - \mathbf{y} \rangle$, we have

$$\left\langle \mathbf{d}^{\mathrm{I}}, f^{(k)}(\mathbf{X}) - \mathbf{y} \right\rangle = -\frac{\eta}{N|\mathcal{C}_k|} \sum_{u=0}^{\tau-1} (f^{(k)}(\mathbf{X}) - \mathbf{y})^\top (\mathbf{\Lambda}^{(k,u)} - \mathbf{\Lambda}^{\perp(k,u)})(f_m^{(k,u)}(\mathbf{X}) - \mathbf{y}). \quad (53)$$

Let $T_1$ and $T_2$ denote the following terms

$$T_1 \triangleq -(f^{(k)}(\mathbf{X}) - \mathbf{y})^\top \mathbf{\Lambda}^{(k,u)} (f_g^{(k,u)}(\mathbf{X}) - \mathbf{y}), \quad (54a)$$

$$T_2 \triangleq (f^{(k)}(\mathbf{X}) - \mathbf{y})^\top \mathbf{\Lambda}^{\perp(k,u)} (f_g^{(k,u)}(\mathbf{X}) - \mathbf{y}), \quad (54b)$$

where $f_g^{(k,u)}(\mathbf{X}) \triangleq [f_1^{(k,u)}(\mathbf{X}_1)^\top, \cdots, f_{|\mathcal{C}_k|}^{(k,u)}(\mathbf{X}_{|\mathcal{C}_k|})^\top]^\top$. We are going to bound $T_1$ and $T_2$ separately. $T_1$ can be written as:

$$T_1 = -(f^{(k)}(\mathbf{X}) - \mathbf{y})^\top (\mathbf{\Lambda}^{(k,u)} - \mathbf{H}^{(0)} + \mathbf{H}^{(0)} - \mathbf{H}^\infty + \mathbf{H}^\infty)(f_g^{(k,u)}(\mathbf{X}) - \mathbf{y}) \quad (55a)$$

$$= -(f^{(k)}(\mathbf{X}) - \mathbf{y})^\top (\mathbf{\Lambda}^{(k,u)} - \mathbf{H}^{(0)})(f_g^{(k,u)}(\mathbf{X}) - \mathbf{y})$$
$$\quad - (f^{(k)}(\mathbf{X}) - \mathbf{y})^\top (\mathbf{H}^{(0)} - \mathbf{H}^\infty)(f_g^{(k,u)}(\mathbf{X}) - \mathbf{y})$$
$$\quad - (f^{(k)}(\mathbf{X}) - \mathbf{y})^\top \mathbf{H}^\infty (f^{(k)}(\mathbf{X}) - \mathbf{y})$$
$$\quad - (f^{(k)}(\mathbf{X}) - \mathbf{y})^\top \mathbf{H}^\infty (f_g^{(k,u)}(\mathbf{X}) - f^{(k)}(\mathbf{X})). \quad (55b)$$

First, we bound the norm of $f_g^{(k,u)}(\mathbf{X}) - \mathbf{y}$. It can be shown that

$$\|f_m^{(k,u)}(\mathbf{X}_m) - \mathbf{y}_m\|_2 = \|f_m^{(k,u)}(\mathbf{X}_m) - f_m^{(k,u-1)}(\mathbf{X}_m) + f_m^{(k,u-1)}(\mathbf{X}_m) - \mathbf{y}_m\|_2 \quad (56a)$$

$$\leqslant \|f_m^{(k,u)}(\mathbf{X}_m) - f_m^{(k,u-1)}(\mathbf{X}_m)\|_2 + \|f_m^{(k,u-1)}(\mathbf{X}_m) - \mathbf{y}_m\|_2 \quad (56b)$$

$$\overset{①}{\leqslant} (1 + \eta)\|f_m^{(k,u-1)}(\mathbf{X}_m) - \mathbf{y}_m\|_2, \quad (56c)$$

where ① holds based on the derivation of (43). Applying (56c) recursively yields

$$\|f_m^{(k,u)}(\mathbf{X}_m) - \mathbf{y}_m\|_2 \leqslant (1 + \eta)^u \|f^{(k)}(\mathbf{X}_m) - \mathbf{y}_m\|_2. \quad (57)$$

The bound for $\|f_g^{(k,u)}(\mathbf{X}) - \mathbf{y}\|_2^2$ can thus be derived as

$$\|f_g^{(k,u)}(\mathbf{X}) - \mathbf{y}\|_2^2 = \sum_{i=1}^N \left[ f_g^{(k,u)}(\mathbf{x}_i) - y_i \right]^2 \quad (58a)$$

$$= \sum_{m \in \mathcal{C}_k} \left\| f_m^{(k,u)}(\mathbf{X}_m) - \mathbf{y}_m \right\|_2^2 \quad (58b)$$

$$\leqslant (1 + \eta)^{2u} \left\| f^{(k)}(\mathbf{X}) - \mathbf{y} \right\|_2^2. \quad (58c)$$

Second, following the steps in Lemma 2, it can be shown that with probability at least $1 - \delta$,

$$\|\mathbf{\Lambda}^{(k,t)} - \mathbf{H}^{(0)}\|_2 \leqslant \frac{2\sqrt{2}NR}{\sqrt{\pi}\delta\alpha}. \quad (59)$$

We also bound the difference between $f_g^{(k,u)}(\mathbf{X})$ and $f^{(k)}(\mathbf{X})$ as follows:

$$\|f_g^{(k,u)}(\mathbf{X}) - f^{(k)}(\mathbf{X})\|_2 \overset{①}{\leqslant} \sum_{v=1}^u \|f_g^{(k,v)}(\mathbf{X}) - f_g^{(k,v-1)}(\mathbf{X})\|_2 \quad (60a)$$

$$\overset{②}{\leqslant} \sum_{v=1}^u \eta \|f_g^{(k,v-1)}(\mathbf{X}) - \mathbf{y}\|_2 \quad (60b)$$

$$\overset{③}{\leqslant} \sum_{v=1}^u \eta(1 + \eta)^{v-1} \|f^{(k)}(\mathbf{X}) - \mathbf{y}\|_2 \quad (60c)$$

$$= [(1 + \eta)^u - 1] \|f^{(k)}(\mathbf{X}) - \mathbf{y}\|_2, \quad (60d)$$

where ① holds due to triangle inequality, ② comes from (43), ③ comes from (58c). Plugging the results from (58c), (59), and (60d) into (55b), we have with probability at least $1 - \delta$,

$$T_1 \leqslant \left[ (1 + \eta)^u \left( \frac{2\sqrt{2}NR}{\sqrt{\pi}\delta\alpha} + N\sqrt{\frac{\ln(2N^2/\delta)}{2n}} + \kappa\lambda_0 \right) - (1 + \kappa)\lambda_0 \right] \|f^{(k)}(\mathbf{X}) - \mathbf{y}\|_2^2, \quad (61)$$

where $\kappa$ is the condition number of the matrix $\mathbf{H}^\infty$. Next, consider the bound for $T_2$. The $\ell_2$ norm of $\mathbf{\Lambda}^{\perp(k,u)}$ can be bounded as

$$\|\mathbf{\Lambda}^{\perp(k,u)}\|_2 \leqslant \|\mathbf{\Lambda}^{\perp(k,u)}\|_{\mathrm{F}} \tag{62a}$$

$$= \left( \sum_{i=1}^N \sum_{m \in \mathcal{C}_k} \sum_{j \in \mathcal{I}_m} \left( \frac{1}{n} \sum_{r \in \mathcal{S}_i} \mathbf{x}_i^\top \mathbf{x}_j \mathbb{1}_{ir}^{(k)} \mathbb{1}_{jmr}^{(k,u)} \right)^2 \right)^{\frac{1}{2}} \tag{62b}$$

$$\leqslant \frac{N}{n} |\mathcal{S}_i| \stackrel{①}{\leqslant} \sqrt{\frac{2}{\pi}} \frac{NR}{\delta\alpha}, \tag{62c}$$

where ① comes from Lemma 1. Therefore, we have with probability at least $1 - \delta$,

$$T_2 \leqslant (1 + \eta)^u \sqrt{\frac{2}{\pi}} \frac{NR}{\delta\alpha} \|f^{(k)}(\mathbf{X}) - \mathbf{y}\|_2^2. \tag{63}$$

Combine the results of (61) and (63):

$$\left\langle \mathbf{d}^{\mathrm{I}}, f^{(k)}(\mathbf{X}) - \mathbf{y} \right\rangle \leqslant \frac{\tau}{|\mathcal{C}_k|} \left[ \left( \eta + \frac{(\tau - 1)}{2}\eta^2 + o(\eta^2) \right) \left( \frac{3\sqrt{2}R}{\sqrt{\pi}\delta\alpha} + \sqrt{\frac{\ln\left(\frac{2N^2}{\delta}\right)}{2n}} + \frac{\kappa\lambda_0}{N} \right) \right. $$
$$\left. - \frac{(1 + \kappa)\eta\lambda_0}{N} \right] \|f^{(k)}(\mathbf{X}) - \mathbf{y}\|_2^2. \tag{64}$$

For the inner product term $\left\langle \mathbf{d}^{\mathrm{II}}, f^{(k)}(\mathbf{X}) - \mathbf{y} \right\rangle$, we first bound $\|\mathbf{d}^{\mathrm{II}}\|_2^2$ with probability at least $1 - \delta$:

$$\|\mathbf{d}^{\mathrm{II}}\|_2^2 = \sum_{i=1}^N \left( \frac{1}{\sqrt{n}} \sum_{r \in \mathcal{S}_i} \left[ c_r \sigma \left( (\mathbf{v}_r^{(k+1)})^\top \mathbf{x}_i \right) - c_r \sigma \left( (\mathbf{v}_r^{(k)})^\top \mathbf{x}_i \right) \right] \right)^2 \tag{65a}$$

$$\leqslant \frac{1}{n} \sum_{i=1}^N |\mathcal{S}_i| \sum_{r \in \mathcal{S}_i} \left( c_r \langle \mathbf{v}_r^{(k+1)} - \mathbf{v}_r^{(k)}, \mathbf{x}_i \rangle \right)^2 \tag{65b}$$

$$\leqslant \frac{1}{n} \sum_{i=1}^N |\mathcal{S}_i| \sum_{r \in \mathcal{S}_i} \left( \frac{\eta c_r}{N\sqrt{n}|\mathcal{C}_k|} \sum_{m \in \mathcal{C}_k} \sum_{u=0}^{\tau-1} \sum_{j \in \mathcal{I}_m} (f_m^{(k,u)}(\mathbf{x}_j) - y_j) \mathbb{1}_{jmr}^{(k,u)} \right)^2 \tag{65c}$$

$$\leqslant \frac{\eta^2}{N^2 n^2 |\mathcal{C}_k|^2} \sum_{i=1}^N |\mathcal{S}_i| \sum_{r \in \mathcal{S}_i} \left( \sum_{m \in \mathcal{C}_k} \sum_{u=0}^{\tau-1} \sum_{j \in \mathcal{I}_m} \left| f_m^{(k,u)}(\mathbf{x}_j) - y_j \right| \right)^2 \tag{65d}$$

$$\leqslant \frac{\eta^2}{N^2 n^2 |\mathcal{C}_k|^2} \sum_{i=1}^N |\mathcal{S}_i| \sum_{r \in \mathcal{S}_i} \left( \sum_{m \in \mathcal{C}_k} \sum_{u=0}^{\tau-1} |\mathcal{I}_m| \left\| f_m^{(k,u)}(\mathbf{X}_m) - \mathbf{y}_m \right\|_2 \right)^2 \tag{65e}$$

$$\stackrel{①}{\leqslant} \frac{\eta^2}{N^2 n^2 |\mathcal{C}_k|^2} \sum_{i=1}^N |\mathcal{S}_i| \sum_{r \in \mathcal{S}_i} \left( \sum_{m \in \mathcal{C}_k} \sum_{u=0}^{\tau-1} (1 + \eta)^u |\mathcal{I}_m| \left\| f^{(k)}(\mathbf{X}_m) - \mathbf{y}_m \right\|_2 \right)^2 \tag{65f}$$

$$\stackrel{②}{\leqslant} \frac{1}{N^2 n^2 |\mathcal{C}_k|^2} \sum_{i=1}^N |\mathcal{S}_i| \sum_{r \in \mathcal{S}_i} \left( \sum_{m \in \mathcal{C}_k} ((1 + \eta)^\tau - 1) |\mathcal{I}_m| \left\| f^{(k)}(\mathbf{X}_m) - \mathbf{y}_m \right\|_1 \right)^2 \tag{65g}$$

$$\stackrel{③}{\leqslant} \frac{1}{N n^2 |\mathcal{C}_k|^2} \sum_{i=1}^N |\mathcal{S}_i| \sum_{r \in \mathcal{S}_i} \left( ((1 + \eta)^\tau - 1) \left\| f^{(k)}(\mathbf{X}) - \mathbf{y} \right\|_2 \right)^2 \tag{65h}$$

$$\stackrel{④}{\leqslant} \frac{2R^2}{\pi\delta^2\alpha^2 |\mathcal{C}_k|^2} \left( \tau\eta + \frac{\tau(\tau - 1)}{2}\eta^2 + o(\eta^2) \right)^2 \|f^{(k)}(\mathbf{X}) - \mathbf{y}\|_2^2. \tag{65i}$$

where ① comes from (57), ② holds due to $\|\mathbf{a}\|_1 \leqslant \|\mathbf{a}\|_2$, ③ holds due to $\|\mathbf{a}\|_1 \leqslant \sqrt{\dim(\mathbf{a})}\|\mathbf{a}\|_2$, ④ is from Lemma 1. With probability at least $1 - \delta$, the inner product term can thus be bounded as

$$\left\langle \mathbf{d}^{\mathrm{II}}, f^{(k)}(\mathbf{X}) - \mathbf{y} \right\rangle \leqslant \frac{\sqrt{2}\tau R}{\sqrt{\pi}\delta\alpha|\mathcal{C}_k|} \left( \eta + \frac{(\tau - 1)}{2}\eta^2 + o(\eta^2) \right) \left\| f^{(k)}(\mathbf{X}) - \mathbf{y} \right\|_2^2. \tag{66}$$

The bound for the difference term is derived as

$$\left\| f^{(k+1)}(\mathbf{X}) - f^{(k)}(\mathbf{X}) \right\|_2^2 \leqslant \sum_{i=1}^{N} \left( \frac{\eta}{\sqrt{n}} \sum_{r=1}^{n} c_r \langle \mathbf{v}_r^{(k+1)} - \mathbf{v}_r^{(k)}, \mathbf{x}_i \rangle \right)^2 \tag{67a}$$

$$\leqslant \frac{1}{|\mathcal{C}_k|^2} \left( \tau\eta + \frac{\tau(\tau - 1)}{2}\eta^2 + o(\eta^2) \right)^2 \left\| f^{(k)}(\mathbf{X}) - \mathbf{y} \right\|_2^2. \tag{67b}$$

Combine the results of (64), (66) and (67b):

$$\left\| f^{(k+1)}(\mathbf{X}) - \mathbf{y} \right\|_2^2 \leqslant \left\{ 1 + \frac{2\eta\tau}{|\mathcal{C}_k|} \left[ \left( \frac{4\sqrt{2}R}{\sqrt{\pi}\delta\alpha} + \sqrt{\frac{\ln\left(\frac{2N^2}{\delta}\right)}{2n}} + \frac{\kappa\lambda_0}{N} \right) \right. \right.$$
$$\left. \left. - \frac{(1 + \kappa)\lambda_0}{N} \right] + \frac{\eta^2\tau^2}{|\mathcal{C}_k|^2} + o(\eta^2) \right\} \left\| f^{(k)}(\mathbf{X}) - \mathbf{y} \right\|_2^2. \tag{68}$$

Let $R = O\left(\frac{\delta\alpha\lambda_0}{N}\right)$, $n = \Omega\left(\frac{N^2}{\lambda_0^2} \ln \frac{N^2}{\delta}\right)$, and $\eta = O\left(\frac{\lambda_0}{\tau N|\mathcal{C}_k|}\right)$, we have

$$\left\| f^{(k+1)}(\mathbf{X}) - \mathbf{y} \right\|_2^2 \leqslant \left( 1 - \frac{\eta\tau\lambda_0}{2N|\mathcal{C}_k|} \right) \left\| f^{(k+1)}(\mathbf{X}) - \mathbf{y} \right\|_2^2. \tag{69a}$$

$\square$

