# OpenReview forum: "Neural Tangent Kernel Empowered Federated Learning"
_ICLR.cc/2022/Conference — ICLR 2022 Submitted_

### Official Review · Reviewer_4JCT · 2021-10-31

**Correctness:** 3
**Technical Novelty And Significance:** 2
**Empirical Novelty And Significance:** 2
**Recommendation:** 5
**Confidence:** 3

**Main Review:**

Strengths:

- The paper provides a novel approach to FL using NTK paradaigm
- The text is well written and clear to follow

Weaknesses:

- The notations can improve to make both the algorithm and the analysis more clear. It seems that $X_m$ and $Y_m$ are used for worker m's samples, and $X^{(k)}$ and $Y^{(k)}$ are used for the sample in round $k$. In the analysis and equation (11) $X$ is used without subscript/superscript. Could you clarify if $X^{(k)}$ and $Y^{(k)}$ are concatenations of $X_m$ and $Y_m$ at round $k$? What do X and y in Theorem 1 represent?

- I may be missing something here, but it seems that the closed form expressions for the trained network using NTK, for example, the ones given in equations (8), (10) and (11) of (Lee et al., 2019) also depend on the inverse kernel matrix. That seems to be standard in kernel methods in general. In fact, kernel methods are known to be computationally prohibitive due to the matrix inverse step. It is not clear to me why and how the closed form solution given in equation (8) does not depend on the kernel inverse. I augest to use clear references for all the results which are not proven in this paper, to make the paper self-contained.
It seems X in equation (8) is only the sample. Has this become possible because (8) does not actually represent a closed form solution, but rather only the output values at the sample?

- While the novel framework is nicely contextualized through a literature review, the information that is uploaded by the workers in this framework violates the privacy preserving feature of FL. Noting that privacy preserving is the main motivation for the FL framework, this can be a problematic aspect of the novel framework. Although CP-NTK-FL is introduced to improve the privacy by compression and shuffling, without a clear discussion and characterization of privacy guarantee, the framework may not be effective. I think authors should include a clear discussion on this issue.

**Summary Of The Paper:**

This paper introduces new algorithms for the federated learning (FL) farmwork using neural tangent kernel (NTK) paradigm. Two algorithms are referred to as NTK-FL and CP-NTK-FL, where the latter is a variant of the former for improved communication efficiency and privacy preserving. The proposed algorithms are aimed to address statistical heterogeneity across the workers. An important point is that unlike typical training algorithms for the FL setting, here the workers upload the labels and sample Jacobian matrices (representing NTK) to the server. The server then uses tools from NTK to obtain the trained neural network (instead of using gradient descent). Theoretical and empirical validations (on MNIST, Fashion-MNIST and EMNIST), and comparisons with FedAvg are provided.

**Summary Of The Review:**

In summary, I appreciate the novel framework introduced in this paper for FL problem. Some of the results are however not completely clear to me. I also suggest a clear discussion on privacy issue that is central to the FL problem.

---

> ### Author Response · Authors · 2021-11-16
> **Response to Reviewer 4JCT Part 1**
>
> 1. > Could you clarify if $X^{(k)}$ and $Y^{(k)}$ are concatenations of $X_m$ and $Y_m$ at round $k$?
>
>     Yes. We clarified it by adding the sentence after Eq (8) (new numbering) in Section 4.1:
>
>     -  $\mathbf{X^{(k)} }= [\mathbf{X_1^\top}, \dots, \mathbf{X^\top_{\mathcal{C}_k}}]^\top$ denotes a concatenation of worker training examples,
>
>     and  $\mathbf{Y^{(k)} }= [\mathbf{Y_1^\top}, \dots, \mathbf{Y^\top_{\mathcal{C}_k}}]^\top$ denotes a concatenation of worker labels.
>
>     > In the analysis and equation (11) $X$ is used without subscript/superscript.
>
>     We thank the reviewer for pointing out the typo in Eq (11) (old numbering, Eq. (8) in the revised version). We added the superscript $k$ to Eq. (8) in the revised version.
>
>     > What do X and y in Theorem 1 represent?
>
>     We define $\mathbf{X}$ as the concatenations of inputs from all clients.
> $\mathbf{y}$ denotes concatenations of all labels; we use small $\mathbf{y}$ because we consider a vector of scalar outputs for simplicity.
> Since it is not time variant, we remove the superscript $k$.
> We moved the definition of $\mathbf{X}$ and $\mathbf{y}$ from the appendix to Section 5 to avoid confusion.
>
> 2. The solution for the neural network function $\mathbf{f}$ does not rely on the matrix inverse,
> e.g., Eq. (9) in (Lee et al., 2019) and Eq. (7a) (new numbering) in this work.
> The weight evolution in Eq. (7b) (new numbering) unfolds the gradient descent step without performing gradient inverse.
> We have explicitly added the sentence in Section 4.1 after Eq (7b) for clarification:
>     - The weight evolution in (7b) is derived by unfolding the gradient descent steps.
>
> 3. We thank the reviewers for the suggestion.
> We revised the second paragraph and last paragraph in Section 4.2 and gave a detailed discussion on confidentiality and membership privacy, in response to your suggestion and Reviewer C84w's suggestion:
>     - For security, we investigate a threat model where a curious server may perform membership inference attacks or data reconstruction attacks. Compared to the averaged gradient, sample-wise Jacobian matrices are more expressive, which may facilitate such attacks from the aggregation server. ... The shuffling step defends against both model inversion and membership inference attacks. Considering the high dimensionality of the neural network weight, the reconstruction attack becomes computationally infeasible. As provable differential privacy guarantees do not explicitly protect against the reconstruction attack (Zhang et al., 2020), we leave a thorough privacy study for future work.
>
> Zhang et al. The Secret Revealer: Generative Model-inversion Attacks Against Deep Neural Networks. CVPR 2020.

---

> > ### Comment · Reviewer_4JCT · 2021-11-21
> > **Feedback after Author Response**
> >
> > Thank you for your response and clarifying the notation and the solution for the neural network function. I find the idea of the paper interesting. I however find the issues around practicality and privacy raised here as well as by other reviewers significant limitations for this work. I maintain my rating.

---

### Official Review · Reviewer_uPVB · 2021-11-02

**Correctness:** 2
**Technical Novelty And Significance:** 3
**Empirical Novelty And Significance:** 1
**Recommendation:** 5
**Confidence:** 4

**Main Review:**

- This paper uses the NTK method as a tool for predicting the output of neural networks. As in eq (6), the dynamics of the gradient flow can be expressed by the neural tangent kernel (NTK). However, the NTK does not change over time under the **infinite-width regime** (please see [1]). The paper does not mention anything about the width limit of networks. Moreover, the neural network used in the proposed FL framework cannot have infinite width, hence, the evolution dynamics (Eq. 8, 10(b)) may be not valid to predict the network outputs.

    [1] Chizat et al., On Lazy Training in Differentiable Programming. NeurIPS. 2019


- The paradigm requires each worker to send a Jacobian tensor with respect to its assigned dataset, which increases communication overhead. Indeed, the transmitted parameters are used for updating the global parameter governed by Eq. (10b). Can the communication bottleneck improve by transmitting only essential factors for the updates? For example, instead of sending J_m^{(k)}, f^{(k)}(X_m), Y_m, one can transmit the corresponding sub-matrix of w^{(k,t)}, which leads to reduce the communication overhead. Is there any reason that workers send the Jacobian matrix?


- CP-FL-NTK variant uses dimension reduction via a random projection with a shared seed. A larger compression ratio (beta) reduces the communication overhead better but the performance also does so. How does degradation relate to the compression ratio? Is there any provable guarantee or practical guidance to choose a proper value of beta? And what is the exact random projection? Is it Gaussian random projection? or random sketching? Moreover, can the corresponding privacy guarantee be analyzed?


- In Remark 1, the authors argue that Eq. (19) implies that convergence in Eq. (17) is faster than that in Eq. (18). But, precisely, it is not true because of $1- t \cdot (\eta \lambda_0 / (2N)) \leq 1- (\eta \lambda_0 / (2N))^t$ for integers $t \geq 1$.

- Minor Issues:

  - The big-O notations usually do not include constants, hence writing $\Omega(N^2 / \lambda^2 * \log(N/d))$ in Theorem 1 seems more natural.

  - It is good to briefly describe FedAvg algorithms for better understanding.

  - A d-dimensional Dirichlet distribution requires a d-dimensional parameter vector (alpha), but it is chosen by scalar values in section 6. Does it mean that the vector with same values? Do the authors have a chance to try other parameter (alpha) settings?

  - It would be great if graph colors and markers in Figures 4 and 5 match each other.


**Summary Of The Paper:**

This paper proposes a federated learning (FL) paradigm empowered by the neural tangent kernel (NTK) framework. The NTK under the infinite-width regime allows us to analyze a dynamic of the corresponding neural network without a gradient descent algorithm. The authors utilize it for predicting the best parameters aggregated from multiple workers. However, this framework requires transmitting Jacobian matrices between workers and the aggregation server, which results in increasing computational overhead and exposing more private information. To this end, they adopt dimensionality reduction via randomness-sharing projection, zeroing out compression as well as shuffling. They also address that their NTK-based FL scheme has a faster convergence rate compared to that of FedAvg, for a two-layer network under specific assumptions. Finally, empirical results support that the proposed FL method performs better than other FL algorithms and achieves similar test accuracy to the ideal centralized case.

**Summary Of The Review:**

The paper is well-written and the methodology is very clear. However, it seems that the paper needs to address the usage of NTK more concretely, e.g., clarifying the width limit. In addition, the second algorithm with communication efficiency and privacy protection also needs more rigorous analyses. Overall, this paper is under the bar of acceptance.

---

> ### Author Response · Authors · 2021-11-16
> **Response to Reviewer uPVB Part 1**
>
> 1. We thank the reviewer for pointing out potential limitations of the NTK framework.
> >  The paper does not mention anything about the width limit of networks.
>
>     In Theorem 1 of Section 5 , we require the neural network width $n=\Omega\left(\frac{N^{2}}{\lambda_{0}^{2}} \ln \frac{N^{2}}{\delta}\right)$. When $\delta$ approaches zero, the width limit approaches infinity.
>
>     >  The neural network used in the proposed FL framework cannot have infinite width, hence, the evolution dynamics (Eq. 8, 10(b)) may be not valid to predict the network outputs.
>
>     Lee et al. (2019) studied  the NTK evolution in the weight space and showed that it is valid for the neural network with a finite width.
> For separate communication rounds, the server selects different sets of clients with different training examples, which changes the loss landscape. In this sense, we have dynamic empirical neural tangent kernels along the communication rounds.
> In Eq. (9) (new numbering), we designed methods to efficiently update the global weight $\mathbf{w^{(k)}}$ rather than purely employing the NTK to predict the neural network output in the function space. The neural network makes prediction based on the updated weight rather than the tangent kernel during training/inference.
>
> Lee et al. Wide Neural Networks of Any Depth Evolve as Linear Models Under Gradient Descent. NeurIPS 2019.
>
> 2.  The Jacobian matrices are more expressive representations of the data compared to aggregated gradients.
> We added a discussion in Section 4.1 to bring out key points:
>     - The update being sent for NTK-FL regarding the $i$th training example of the $m$th worker for NTK-FL is $ \mathbf{J_m} = [\nabla \mathbf{f_1}(\mathbf{x_{m,i}}), \dots, \nabla \mathbf{f_{d_2}}(\mathbf{x_{m,i}})]^\top$,
> whereas the gradient update being sent for FedAvg is $\nabla L(\mathbf{w}; \mathbf{x_{m,i}}, \mathbf{y_{m,i}}) =  \frac{1}{d_2}\sum_{j=1}^{d_{2}}\left(\hat{y_{m, i, j}}-y_{m, i, j}\right) \nabla \mathbf{f_j}(\mathbf{x_{m, i}})$, a weighted sum of coordinates of $\mathbf{J_m}$.
> By sending Jacobian matrices $\mathbf{J}_m$ and jointly processing them on the server, NTK-FL delays the more aggressive data aggregation step after the communication stage and therefore better approximates the centralized learning setting than FedAvg does.
>
> 3. > How does degradation relate to the compression ratio?
>
>     In Figure 6, we empirically verified the impact of $\beta$ on performance degradation.
> Since $N_{m}^{\prime}=\beta N_{m}$, $\beta$ is also the scaling factor on the compression ratio.
> For example, $\beta=0.5$ reduces the overhead by a factor of $2$.
> In Figure 6, we showed that as long as $\beta$ and $d'_1$ are not too small, the algorithm is robust to different hyperparameter settings.
>
>     > Is there any provable guarantee or practical guidance to choose a proper value of $\beta$?
>
>     According to the probably approximately correct (PAC) framework, using fewer training examples always harm the model generalizability.
> $\beta$ serves as a tradeoff between the model performance and the communication overhead.
> In practice, we suggest clients choose $\beta$ based on the available resources, e.g., choosing a larger $\beta$ if they have enough bandwidths.
>
>     > And what is the exact random projection? Is it Gaussian random projection? or random sketching
>
>     It is not random sketching. The mathematical expression is the same with the Gaussian random projection, except that we do not initialize the random matrix $\mathbf{P}$ with orthogonal column vectors. We revised the descriptions in Section 4.2 "Jacobian Dimension Reduction Paragraph":
>     - $\mathbf{P}$ is a projection matrix generated based on a seed $\rho$ with IID standard Gaussian entries.
>
>     > Moreover, can the corresponding privacy guarantee be analyzed?
>
>     Our methods defend against the data reconstruction attack performed by a curious server. We added the following sentence in Section 4.2:
>
>     - As provable differential privacy guarantees do not explicitly protect against the reconstruction attack (Zhang et al., 2020), we
> leave a thorough privacy study for future work.
>
> Zhang et al. The Secret Revealer: Generative Model-inversion Attacks Against Deep Neural Networks. CVPR 2020.
>
> 4. We thank the reviewer for pointing it out.
> We modified Remark 1 to
>     - we compare the Binomial approximation of the decay coefficient in Theorem 1 with the
> decay coefficient in Theorem 2, i.e., $1-\frac{\eta_{1} t^{(k)} \lambda_{0}}{2 N} + O(\eta_1^2) < 1-\frac{\eta_{2} \tau \lambda_{0}}{2 N\left|\mathcal{C}_{k}\right|}$, for $\eta_1 \ll 1$. The number of training examples $N$ is often larger than $t^{(k)}$.
> For example, if we have $100$ clients, each of which has more than $100$ data points, then $N$ is on the order of $10^4$.
> Considering the choice of the learning rate $\eta_1$, the Binomial approximation holds in this inequality.

---

> > ### Author Response · Authors · 2021-11-16
> > **Response to Reviewer uPVB Part 2**
> >
> > > The big-O notations usually do not include constants, hence writing $\Omega\left(N^{2} / \lambda^{2} * \log (N / d)\right)$  in Theorem 1 seems more natural.
> >
> > We thank the reviewer for the suggestion.
> > We have made the change accordingly.
> >
> > > It is good to briefly describe FedAvg algorithms for better understanding.
> >
> > We thank the reviewer for the suggestion.
> > We added descriptions of FedAvg in the Introduction:
> >     - In FedAvg, workers perform stochastic gradient descent (SGD) to update the local models and upload the weight vectors to the server.
> > 	A new global model is constructed on the server by averaging the received weight vectors.
> >
> > >  A d-dimensional Dirichlet distribution requires a d-dimensional parameter vector (alpha), but it is chosen by scalar values in section 6.
> > 	Does it mean that the vector with same values?
> >
> > Yes. We followed the notation of the symmetric Dirichlet distribution in which all parameters are equal [1].
> >
> > [1] Good et al. On the Application of Symmetric Dirichlet Distributions and Their Mixtures to Contingency Tables. The Annals of Statistics. 1976.
> >
> > >	Do the authors have a chance to try other parameter (alpha) settings?
> >
> > We didn't use a vector alpha, because the symmetric Dirichlet distribution can provide a smooth simulation for different degrees of heterogeneity in FL according to the literature [2-4].
> >
> > [2] Hsu et al. ``Measuring the Effects of Non-identical Data Distribution for Federated Visual Classification.'' arXiv 2019.
> >
> > [3] Chen et al. ``FedBE: Making Bayesian Model Ensemble Applicable to Federated Learning.'' In ICLR 2021.
> >
> > [4] Yoon et al. ``FedMix: Approximation of Mixup Under Mean Augmented Federated Learning''. In ICLR 2021.
> >
> > > It would be great if graph colors and markers in Figures 4 and 5 match each other.
> >
> > We thank the reviewer for the suggestion.
> > We changed the colors and markers in the revised version.

---

> > > ### Comment · Reviewer_uPVB · 2021-11-22
> > > **After the author response**
> > >
> > > Thank you to the authors for taking the time to respond to my review, and for the clarifications to my understanding. It is satisfactory that the authors updated all minor concerns to their updated paper. However, the main contribution is still limited, and remark 1 (which argues that the proposed method converges faster than FedAvg) depends on the condition of learning rate and is not always true. Hence, I maintain my score.

---

### Official Review · Reviewer_KsCY · 2021-11-03

**Correctness:** 3
**Technical Novelty And Significance:** 2
**Empirical Novelty And Significance:** 2
**Recommendation:** 5
**Confidence:** 3

**Main Review:**

Strengths:
+ The idea of using NTK for model optimization without gradient descent in the FL paradigm is interesting and somewhat novel.
+ Inherent issues such as communication cost and data privacy of the proposed method are properly discussed.

Weaknesses:

A. Method
- In a way, the proposed method shifts much of the computation to the server-side (e.g., weight update and selecting the best one). The various integer update steps for the weight evolution are not clearly specified in the experiments (reproducibility). And the time cost for the grid search of $t^{(k)}$ should be discussed.
- To address the risk of data leakage and the high communication cost of FL-NTK, random data projection, and Jacobian shuffling are used. Although it seems that these tricks will do the job, additional resources (i.e., the trusted key server and the shuffling server) are required, which may not be available for some applications.

B. Experiment evaluation
- The network used in the experiments, i.e., a multilayer perception with 100 hidden nodes, is too simple. More standard network architectures like ResNets (e.g., ResNet-18) should be employed for evaluation and comparison.
- The datasets used in the experiments are also small, and not diverse (basically MNIST and its variants). It would be more convincing to show results and comparison on larger datasets (more samples and classes), such as CIFAR-100 and Tiny-ImageNet.
- In the data heterogeneity experiments, a few popular baselines such as FedProx and SCAFFOLD are not compared.
- In Fig. 5, the performance of FL-NTK does not increase as the level of data non-IIDness goes down. An accuracy drop is observed when $\alpha=0.3$. Can you provide an explanation for this?
- For CP-NTK-FL, communication with the trusted key server and the shuffling server is also required, in addition to the communication between clients and server. Therefore, a breakdown of the comm. cost is helpful information.

**Summary Of The Paper:**

This paper presents a new federated learning (FL) framework that employs the neural tangent kernel (NTK) method to replace the widely used gradient descent algorithm for optimization. To improve communication efficiency and privacy-preserving features, data sampling and random projection techniques are used in the proposed FL-NTK framework. Experiments are conducted to demonstrate the advantages of the proposed FL-NTK in the robustness to data heterogeneity and communication efficiency, as compared to the baseline FedAvg.

**Summary Of The Review:**

Overall, the proposed NTK approach for FL is interesting and has novelty. However, there are a few key limitations/weaknesses: (1) FL-NTK has a higher risk of data leakage and communication cost compared to FedAvg. The proposed CP-FL-NTK may address these issues but at the cost of requiring additional resources (i.e., the trusted key server and the shuffling server), which could limit the practical usage of the method. (2) As a general approach, a more comprehensive and rigorous experimental evaluation is expected to fully demonstrate the effectiveness of the proposed method.

---

> ### Author Response · Authors · 2021-11-16
> **Response to Reviewer KsCY Part 1**
>
> 1. We thank the reviewer for pointing out the potential concern of choosing the local steps. In Remark 1, we say that $t^{(k)}$ is on the order of $10^2$ to $10^3$. For reproducibility, the implementation is available at https://anonymous.4open.science/r/ntk-fed-79F3/. We added the experiment setting in the revised paper:
>
>     - The number of update steps $t^{(k)}$ is chosen over the grids {$100, 200, \dots, 2000\$} .
>
>     For the time cost, we added:
>     - Based on the closed-form solution in (10b), the  search of $t^{(k)}$ over the grid {$t_1, \dots, t_\Psi$} can be completed in $O(\Psi)$ time.
>
> 2. We modified Section 4.2 and explicitly mentioned that the data leakage risk comes from the server membership inference and data reconstruction attacks. We agree that the trusted key server and the shuffling server require extra resources and can be burdens in certain applications. However, we do want to emphasize that shuffling server and trusted key server are optional building modules to enhance confidentiality and membership privacy. The base version with the compression of the Jacobian matrices is capable of defending against the data reconstruction attacks, which has been empirically verified (Zhu et al., 2019).
>
> Zhu et al. Deep Leakage From Gradients. NeurIPS 2019.
>
> 3. We thank the reviewer for pointing out the issue of model size.
> We use this multilayer perceptron based on the following three considerations:
> (1). We want to avoid the mismatch between the analysis and the experimental verification.
> (2). NTK modeling for other neural network structures such as CNNs is still an open research topic.
> For example, Lee et al. (2020) pointed out that NTK cannot precisely model CNNs.
> (3). NTK has difficulties on larger models without further engineering optimizations at the current stage, including memory issues and modeling various network layers.
> We are looking forward to obtaining more empirical results in future work.
>
> 4. We thank the reviewer for pointing out the limitation of datasets used in the original submission.
> In the revised manuscript, we added the experiments on the CIFAR-10 dataset in Appendix A, which goes beyond the MNIST datasets.
> We had tried to evaluate the method on CIFAR-100 and ImageNet,
> but we found that (1) with a relatively small neural network, the test accuracy was poor; and
> (2) with a relatively large neural network, the current implementation needed nontrivial improvement such as sophisticated parallelization and resolving the memory issue. In this work, we focus on building the first proof-of-concept prototype that uses the NTK framework to replace client local update in FL, and we are looking forward to extending the paradigm to more sophisticated datasets.
>
> 5. (1) Since FedNova consistently outperforms FedProx (reported by Wang et al. 2020), we choose to report the FedNova results as a stronger baseline.
> (2) Scaffold mainly focuses on the cross-silo setting by maintaining the state variables on each client.
> In our study, we consider a general cross-device setting where the sever samples stateless clients in each round.
> In this cross-device setting, Scaffold does not outperform FedAvg (reported by Reddi et al. 2021).
> Because the Scaffold performance is not consistently satisfactory and is worse than FedAvg, we do not report the Scaffold results.
>
> Wang et al. Tackling the Objective Inconsistency Problem in Heterogeneous Federated Optimization. NeurIPS 2020.
>
> Reddi et al. Adaptive Federated Optimization. ICLR 2021.
>
> 6. In Fig. 5, the performance of DataShare does not always improve as the degree of heterogeneity reduces, which shares the same pattern with NTK-FL.
> NTK-FL better approximates the centralized learning setting than FedAvg by sharing more expressive data representations.
> In this sense, NTK-FL resembles the DataShare baseline.
> The performance of NTK-FL and DataShare has little dependence on the degree of heterogeneity;
> the bottleneck in these methods is more on other factors, such as the number of clients sampled in each round.
>
> 7. We thank the reviewer for the suggestion, but as the uplink communication cost is the main focus in our study, we believe that the breakdown of the communication cost is not necessary.
> Except the negligible handshake overhead, the calculation of uplink communication cost remains unchanged when introducing the trusted key server and the shuffling server.
> In the second paragraph of Section 4, we added:
>
>      - For communication, we follow the widely adopted analysis framework in wireless communication to examine only the client uplink overhead, assuming that the downlink bandwidth is much larger and the server will have enough transmission power (Tran et al., 2019).
>
> Tran et al. Federated Learning Over Wireless Networks: Optimization Model Design and Analysis. INFOCOM 2019.

---

> ### Comment · Area_Chair_P5f1 · 2021-11-28
> **Please provide feedback**
>
> Dear Reviewer KsCY,
>
> The discussion deadline is only two days away, and we need to reach a consensus soon to recommend a decision. Please go over the responses from the authors on your comments, and revise your rating or review when necessary.
>
> Thanks,
> Area Chair

---

### Official Review · Reviewer_C84w · 2021-11-03

**Correctness:** 3
**Technical Novelty And Significance:** 2
**Empirical Novelty And Significance:** 3
**Recommendation:** 5
**Confidence:** 4

**Main Review:**

This paper combines two popular relevant machine learning frameworks, NTK and federated learning, with the aim of addressing statistical heterogeneity in federated settings. The paper proposes a practical implementation that addresses potential communication burdens. The experimental section provides interesting experimentation showing this framework outperforms FedNova and FedAvg on three datasets.

The paper is well organized and clear. Previous work is discussed and necessary background is introduced. The experimental section also presents interesting results. Plots and experiment details are discussed and explained.

Weaknesses / clarifications.
- My main criticism is that throughout the paper it is mentioned that the NTK “inherently solves the non-IID data problem”, when learning global model under statistical heterogeneity of workers. Yet, from the formulation of the NTK this is not evident and it is not clearly developed in the paper, for example by  explaining why the Jacobian enhances generalization while gradients don’t. Why is this framework more robust? I acknowledge that at the end of section 4.1 there is a paragraph on this, however it does not respond to my question as it affirms that the global kernel $H^{(k)}$ is more expressive than gradients without an explanation of why.

- It is not very clear for me how this algorithm is different from Algorithm 1 in [HLSY21]. Further, it would also help to differentiate the paper from this previous to include a discussion on how the presented convergence results are different from  the ones in [HLSY21].

- Given that several $t^{k}$ are tested at each round, how does this affect the convergence result?


- Throughout the paper a”privacy” is used in different contexts. I would suggest a clarification of what privacy means in each context and what attacks are prevented by taking a specific action. For example, the random projections do not protect from an "honest but curious" server, since all workers share the same projection matrix. This step does not provide Differential privacy. Some differential privacy guarantees can be provided from the shuffling model, but again this is not very clearly explained, and what it is protecting from.


Minor:
- I understand NTK intrinsic difficulties to try on larger models, but given that the contribution of this paper is a practical algorithm it would be interesting to see results in more realistic FL settings, perhaps a small language task.
- Given that FL is an applied field, It would be good to have a small discussion on practical aspects of FL like what to expect under user intermittency, privacy (see above), computation and memory costs, and formal communication costs, e.g. $O(poly(M, N_m, \beta, ...))$ and how it compares with SotA algorithms.


**Summary Of The Paper:**

This paper proposes an algorithm for federated learning that leverages recent advances in the NTK framework. Participants share sample-wise jacobian matrices instead of model weights or gradients, based on the intuition that the NTK is able to capture useful statistical information when learning under statistical heterogeneity. The paper presents numerical results showing that the proposed approach maintains efficiency across different heterogeneity levels. Further, the authors introduce a practical implementation that reduces communication, and compare this approach with FedAvg and FedNova.

**Summary Of The Review:**

This paper proposes a practical algorithm for FL based on recent advances on the NTK, and FL-NTK. It provides interesting preliminary and promising results. However, there are several aspects related to clarifying intuitions and differentiating this paper from previous work that still need to be addressed.

---

> ### Author Response · Authors · 2021-11-16
> **Response to Reviewer C84w Part 1**
>
> 1. We thank the reviewer for pointing out this issue.
> The Jacobian tensor is an unaggregated version of gradients and preserves more information.
> We added a discussion to the last paragraph of Section 4.1:
>
>     - The update being sent for NTK-FL regarding the $i$th training example of the $m$th worker for NTK-FL is $\mathbf{J_m} = [\nabla \mathbf{f_1}(\mathbf{x_{m,i}}), \dots, \nabla \mathbf{f_{d_2}}(\mathbf{x_{m,i}})]^\top$, whereas the gradient update being sent for FedAvg is $ L(\mathbf{w}; \mathbf{x_{m,i}}, \mathbf{y_{m,i}}) =  \frac{1}{d_2}\sum_{j=1}^{d_{2}}\left(\hat{y_{m, i, j}}-y_{m, i, j}\right) \nabla \mathbf{f_{j}}(\mathbf{x_{m, i}})$, a weighted sum of coordinates of $\mathbf{J_m}$. By sending Jacobian matrices $\mathbf{J}_m$ and jointly processing them on the server, NTK-FL delays the more aggressive data aggregation step after the communication stage and therefore better approximates the centralized learning setting than FedAvg does.
>
> 2. > It is not very clear for me how this algorithm is different from Algorithm 1 in [HLSY21].
>
>     We added a paragraph in Section 4 to clarify the differences:
>     - Algorithm 1 [HLSY21] is FedAvg, where clients perform SGD and transmit gradients to the server to update a two-layer neural network. In our proposed NTK-FL, each client transmits Jacobian matrices without performing local SGD steps. The model weight is updated via NTK evolution in Eq. (10b) of the original manuscript or Eq. (7b) of the revised manuscript. The main differences include: 1. clients transmit more expressive Jacobian matrices to improve model performance in the non-IID FL setting; 2. the computation is shifted to the server.
>
>     > Further, it would also help to differentiate the paper from ... the ones in [HLSY21].
>
>     In Section 2, we said that "our work does not focus on pure convergence analyses of existing algorithms." In Section 5, we compared the convergence rate of NTK-FL and FedAvg. Except technical details, we did not improve the FedAvg convergence results of [HLSY21].
>
> 3. In each communication round, we evaluate the model with {$t_1, \dots, t_n$} and select the one giving the lowest loss. In theory, the selection of $t^{(k)}$ minimizes the loss $L$ in (12a) and accelerates the convergence as much as possible. In practice, the weight evolution will diverge from the theoretical trajectory for a neural network with a finite depth when the number of update steps increases [1]. As a result, the training loss will be higher than the theoretical loss.
>
>     [1] Lee et al. Finite Versus Infinite Neural Networks: An Empirical Study. NeurIPS 2020.
>
> 4. We thank the reviewer for raising this concern. We revised the second paragraph and the last paragraph in Section 4.2:
>     - For security, we investigate a threat model where a curious server may perform membership inference attacks or data reconstruction attacks. Compared to the averaged gradient, sample-wise Jacobian matrices are more expressive, which may facilitate such attacks from the aggregation server. ... The shuffling step defends against both model inversion and membership inference attacks.
> Considering the high dimensionality of the neural network weight, the reconstruction attack becomes computationally infeasible.
> As provable differential privacy guarantees do not explicitly protect against the reconstruction attack [2], we leave a thorough privacy study for future work.
>
>     > the random projections do not protect from an honest but curious server
>
>     The projection seed is shared by the trusted key server, which follows the concept of trusted third party (TTP) in cryptography.
> We assume the trusted server will not be compromised [3]. For example, the certificate authority (CA) issuing digital certificates in the Transport Layer Security (TLS) protocol is an instance of the trusted third party.
>
>   [2] Zhang et al. The Secret Revealer: Generative Model-inversion Attacks Against Deep Neural Networks. CVPR 2020.
>
>   [3] Van Oorschot. Computer Security and the Internet: Tools and Jewels. Springer 2020.
>
>
> 5. We thank the reviewer for the advice. We had tried language tasks, but we found that extending the proposed method to LSTMs/RNNs needs nontrivial algorithm design efforts. We are looking forward to explorations in future work.
>
> 6. For user intermittency, we allow a low participant rate, the same as the design in FedAvg. For space complexity (memory costs) and uplink communication cost, they are dominated by the third-order tensor. In particular, the communication cost for FedAvg is $O(d)$, and the cost for NTK-FL is $O(N_m d_2 d)$. For the computational cost, we added a discussion to the first paragraph of Section 4.2:
>
>     - Compared to FedAvg, NTK-FL does not incur additional client computational overhead since calculating the Jacobian tensor enjoys the same communication efficiency with computing aggregated gradients. Without locally updating weight vectors, NTK-FL is faster than FedAvg on the client side.

---

> > ### Comment · Reviewer_C84w · 2021-11-21
> > **Thanks for the response.**
> >
> > I thank the authors for the response and the efforts in the updated draft. Some of my queries were clarified. I think this is a very interesting direction to continue exploring.
> >
> >
> > However, unfortunately, I still think more needs to be done to explain the significance of this work. The main contribution is a "practical algorithm" but there is still a big gap between the intuitions (better generalization due to sharing more information rather than just gradients), the theory (how do several hyperparameters affect convergence, the convergence bound still assumes a given $t^k$ but in practice several are tested), and practice (the proposed model is only practical for small tasks, thus the practicality of the algorithm is still questionable).
> >
> > For the above reasons I am maintaining my original score.

---

### Comment · Area_Chair_P5f1 · 2021-11-20
**Rebuttal deadline approaching soon**

Dear Reviewers,

Could you please go over the responses from the authors and provide feedback? The rebuttal deadline is approaching soon (November 22nd) and the authors cannot edit the paper after the deadline.

Thanks,
Area Chair.

---

### Decision · Program_Chairs · 2022-01-20

**Decision:**

Reject

**Comment:**

This paper proposes a novel Federated Learning (FL) framework that leverages the Neural Tangent Kernel (NTK), to replace the gradient-descent algorithm for optimization. Specifically, the workers upload the labels and the Jacobian matrices to the server, and the server uses the tools from the NTK to obtain a trained neural network. However since this could lead to increased communication cost and compromise of data privacy, the authors propose data sampling and random projection techniques to alleviate the problem. The authors provide a theoretical analysis that the proposed scheme has a faster convergence than FedAvg under specific assumptions, and experimentally validate that it significantly outperforms previous FL algorithms, achieving similar test accuracy to ideal centralized cases.

Pros
- The idea of using NTK for model optimization without gradient descent and use of it in the FL setting is both interesting and novel.
- The paper properly discusses and tackles the new challenges posed by the introduction of the new method.
- The paper is well-organized and clearly written, with sufficient discussion of related works and backgrounds.

Cons

- The proposed method puts heavy computational burdens on the server-side.
- The method violates the privacy preserving feature of FL by its nature, and while the proposed compression shuffling alleviates the concern, more discussion is necessary.
- Missing comparison against popular baselines such as FedProx and SCAFFOLD.
- The faster convergence of the proposed method in comparison to FedAvg depends on the learning rate and is not always true.
- There is a gap between the theory and practice, which makes the practicality of the algorithm still questionable.

Although the reviewers found the idea as novel, the proposed techniques for alleviating communication cost and privacy concerns convincing, and considered both the theoretical analysis and experimental validation thorough, all reviewers leaned toward rejection due to critical concerns unanswered. During the discussion period, the authors alleviate many of the minor concerns from the reviewers, but there were still remaining concerns on the gap between the theory and practice on its convergence behavior, and insufficient discussion of the privacy-preserving feature of the proposed method, as well as shifting of computation burdens to the server. Thus, the reviewers reached a consensus that the paper is not yet ready for publication.

Despite the low average score, the novelty of the idea and the quality of the paper is much higher than those of the accepted papers in my batch, and I strongly believe that this will become a high impact paper, if remaining concerns from the reviewers are properly resolved.